# TracrRNA reprogramming enables direct PAM-independent detection of RNA with diverse DNA-targeting Cas12 nucleases

Chunlei Jiao [1], Natalia L. Peeck [1,3], Jiaqi Yu [1,3], Mohammad Ghaem Maghami[1], Sarah Kono[1], Daphne Collias[1], Sandra L. Martinez Diaz [1], Rachael Larose[1] & Chase L. Beisel [1,2] ✉

Many CRISPR-Cas immune systems generate guide (g)RNAs using trans-activating CRISPR RNAs (tracrRNAs). Recent work revealed that Cas9 tracrRNAs could be reprogrammed to convert any RNA-of-interest into a gRNA, linking the RNA's presence to Cas9-mediated cleavage of double-stranded (ds) DNA. Here, we reprogram tracrRNAs from diverse Cas12 nucleases, linking the presence of an RNA-of-interest to dsDNA cleavage and subsequent collateral single-stranded DNA cleavage—all without the RNA necessarily encoding a protospacer-adjacent motif (PAM). After elucidating nuclease-specific design rules, we demonstrate PAM-independent RNA detection with Cas12b, Cas12e, and Cas12f nucleases. Furthermore, rationally truncating the dsDNA target boosts collateral cleavage activity, while the absence of a gRNA reduces background collateral activity and enhances sensitivity. Finally, we apply this platform to detect 16 S rRNA sequences from five different bacterial pathogens using a universal reprogrammed tracrRNA. These findings extend tracrRNA reprogramming to diverse dsDNA-targeting Cas12 nucleases, expanding the flexibility and versatility of CRISPR-based RNA detection.

CRISPR (Clustered Regularly Interspaced Short Palindromic Repeats)-based diagnostics are gaining popularity as fast, affordable, and portable alternatives to traditional PCR-based assays[1]. The most widely adopted formats have involved the single-effector CRISPR-associated (Cas) nucleases Cas12a and Cas13 because of their respective ability to collaterally cleave non-specific single-stranded (ss)DNA or ssRNA reporters upon guide RNA (gRNA)-directed recognition of a target double-stranded (ds)DNA or ssRNA sequence[2–4]. With the discovery of a wide assortment of Cas12 orthologs now grouped into over a dozen sub-types (V-A through V-N)[5–8], many of these nucleases have been co-opted for nucleic-acid detection[2,9–12]. A few of these nucleases were shown to recognize RNA, either as their sole nucleic-acid target (Cas12a2, Cas12g) with the requirement for an adjacent protospacer-flanking sequence[11,13–15], or as one of numerous nucleic-acid targets (Cas12f1)[16]. However, the vast majority of these nucleases require

dsDNA targets to elicit collateral cleavage activity. In turn, dsDNA detection technologies based on Cas12 nucleases have been a major focus for numerous applications ranging from disease diagnosis to food quality control and the detection of environmental pollutants[1,17,18].

Despite the prominence of dsDNA-targeting Cas12 nucleases for nucleic-acid detection, they generally come with two notable restrictions: a strict requirement for a flanking protospacer-adjacent motif (PAM) and limitation to dsDNA targets. These limitations hamper the detection of confined sequence differences, such as SNPs or hypervariable regions in rRNA[19]. The inability to directly detect RNA affects RNA biomarkers such as RNA viruses or alternative splice products, although the common use of reverse transcription and pre-amplification lessens this restriction. Nonetheless, these restrictions have driven numerous efforts to find workarounds. Tackling the first

[1]Helmholtz Institute for RNA-based Infection Research (HIRI), Helmholtz-Centre for Infection Research (HZI), Würzburg, Germany. [2]Medical Faculty, University of Würzburg, Würzburg, Germany. [3]These authors contributed equally: Natalia L. Peeck, Jiaqi Yu. ✉e-mail: chase.beisel@helmholtz-hiri.de

restriction, multiple studies have recently reported PAM-independent detection of dsDNA in which the detected target sequence was not flanked by a PAM. PAM-independent detection was achieved by generating ssDNA targets[20–22], dsDNA targets with a defined single-stranded overhang[23], or dsDNA targets flanked by PAMs as part of DNA amplification[24–27]. Tackling both restrictions, one recent study reported PAM-independent RNA detection with Cas12a by providing a truncated PAM-flanking dsDNA that hybridizes with the PAM-proximal end of the gRNA guide, leaving the PAM-distal end of the gRNA guide available for RNA recognition[28]. While an innovative use of splitting the target to detect RNA, 3′ ends of target RNAs, which create a continuous DNA-RNA target, could only be sensitively detected. One recent study reported direct detection of RNA using Cas12a[29], although nuclease activation by RNA targets has been previously shown to be extremely weak compared to ssDNA or dsDNA targets[13]. Therefore, Cas12 nucleases remain to be fully leveraged for direct, PAM-independent detection of RNA.

One distinct yet unexplored strategy involves tracrRNAs (trans-activating CRISPR RNAs)[30]. tracrRNAs play an essential role in gRNA biogenesis as part of type II CRISPR-Cas systems, the source of Cas9 nucleases, and most type V CRISPR-Cas systems, the source of Cas12 nucleases[5]. In all cases, the tracrRNA hybridizes to the conserved repeat region in CRISPR RNAs (crRNAs) associated with CRISPR-Cas systems[30]. The formed RNA duplex then undergoes processing typically by the endoribonuclease RNase III and complexes with the cognate Cas nuclease. The bound RNA duplex then functions as a gRNA that directs the nuclease to bind and cleave target dsDNA.

Our previous work revealed that Cas9 tracrRNAs can also hybridize to certain cellular RNAs harboring a repeat-like sequence, converting that region into a non-canonical (n)crRNA[31]. The ncrRNA hybridized with the tracrRNA could then be used by Cas9 as a gRNA[31]. In line with this discovery, we and others reprogrammed the hybridizing region of the Cas9 tracrRNA to match to a specific RNA-of-interest that preserves the structure and necessary sequence features of the natural RNA duplex[31–33]. The region of the RNA-of-interest then becomes a gRNA that directs sequence-specific DNA targeting by Cas9. This RNA-sensing capability of reprogrammed tracrRNAs (Rptrs) was the foundation for the in vitro multiplexed RNA detection platform LEOPARD[31], the in vivo transcriptional sensor AGATHA[33] and the in vivo single-cell RNA recording platform TIGER[32]. A PAM was always appended to the target sequence; accordingly, the Cas9 nuclease's particular PAM could be disregarded when selecting the region of an RNA to sense. These advances create the opportunity to extend tracrRNA engineering to Cas12, combining PAM-independent RNA detection previously associated with Cas9 nucleases and collateral cleavage as well as other features associated with the diverse set of tracrRNA-dependent Cas12 nucleases.

Here, we report a distinct strategy to achieve the direct detection of RNA lacking a PAM sequence by reprogramming tracrRNAs associated with diverse Cas12 nucleases (Fig. 1a). We call this approach PUMA, for Programmable tracrRNAs Unlock protospacer-adjacent Motif-independent detection of ribonucleic Acids by Cas12 nucleases. Under this approach, the PAM is appended to a provided dsDNA, allowing any Cas12 nuclease to be used without its associated PAM appearing in the sensed RNA sequence. We first demonstrate the reprogrammability of tracrRNAs associated with Cas12b, Cas12e and Cas12f1 nucleases. Subsequently, we find that RNA detection could be enhanced by using shorter and processed DNA targets. We further show that the absence of a gRNA normally present for standard Cas12-based dsDNA detection assays reduces background collateral activity, lending to RNA detection using a Rptr being more sensitive than DNA detection using an sgRNA. Finally, we demonstrate the utility of PUMA by detecting and differentiating five different bacterial pathogens using a single universal Rptr. Our tracrRNA reprogramming strategy provides a streamlined solution to

detection challenges faced by Cas12 and could further advance CRISPR-based diagnostics at large.

## Results

### Cas12 nucleases offer diverse yet complex opportunities for tracrRNA reprogramming

Type V CRISPR-Cas systems comprise numerous systems that involve tracrRNAs that could be amenable to tracrRNA engineering. Of the 14 subtypes of type V systems defined to-date, eight (associated with Cas12b, Cas12c, Cas12d, Cas12e, Cas12f1, Cas12g, Cas12k and Cas12l) exclusively rely on a tracrRNA for gRNA biogenesis (Fig. 1b)[5,6,11,34–43]. For the remaining systems, the Cas12 nuclease directly recognizes and processes the transcribed repeat, as commonly demonstrated for Type V-A systems and its Cas12a nuclease[7,11,44–47]. Apart from collateral cleavage activity[9,12,35,38,48], some or all of these DNA-targeting nucleases possess features distinct from more traditional Cas9 nucleases, such as pre-crRNA processing[38,49,50], compact nucleases[39–42,48,51], higher optimal temperatures[9,36,52], crRNA-guided transposition[37,53] and T-rich and C-rich PAM recognition[6,11,43,49].

TracrRNA reprogramming involves engineering the anti-repeat region to hybridize with an RNA-of-interest while maintaining the essential sequence and structural features of the natural repeat/anti-repeat (R/AR) duplex recognized by the Cas nuclease. While Cas9-associated RNA duplexes form a simple 25–40 bp stem typically interrupted by a small bulge[54–56], Cas12-associated RNA duplexes adopt distinct and more complicated conformations (Fig. 1c). In addition to a long repeat/anti-repeat (LR/AR) stem often containing an intervening bulge, the reported duplexes associated with Cas12b, Cas12f1, Cas12g and Cas12l also possess a pseudoknot that includes a 5–7 bp short repeat/anti-repeat (SR/AR) stem[6,15,43,57–60]. For Cas12e and Cas12k, the reported RNA duplexes possess a 3-bp triple helix formed by two portions of the anti-repeat sandwiching the repeat in addition to the bulged LR/AR stem[35,53,61,62] (Fig. 1c and Supplementary Fig. 1). Finally, for Cas12c, the reported RNA duplexes form three 4-7 bp disjoint R/AR stems[49] (Fig. 1c). Given the diversity and complexity of these RNA duplexes, we explored the extent to which the RNA duplexes associated with these diverse tracrRNAs can be reprogrammed for RNA detection.

### Reprogrammed tracrRNAs link RNA sensing and dsDNA targeting by Cas12b

We started with the *Bacillus hisashii* Cas12b (BhCas12b) due to the relative simplicity of its RNA duplex comprising a 30-bp LR/AR stem with an intervening bulge, and a 5-bp SR/AR duplex between the LR/AR and the guide[36,57] (Fig. 2a). We sought to investigate the reprogrammability of both stems using a Cas12 cleavage assay conducted with a cell-free transcription-translation (TXTL) system[63]. As part of the assay, purified BhCas12b protein, a gRNA-expressing plasmid and a plasmid encoding the PAM-flanked dsDNA target upstream of a GFP reporter construct were added to a reaction, and we monitored fluorescence over time. Cleavage of the reporter construct leads to loss of GFP expression through rapid degradation of the linearized DNA (Fig. 2b).

To interrogate the reprogrammability of the crRNA-tracrRNA duplex, we began with the intervening bulge in the crRNA-tracrRNA duplex followed by the two stems. Previous studies showed that a bulge in the LR/AR duplex is necessary to maintain the dsDNA targeting activity for SpyCas9 and Sth1Cas9[55]. However, removing this bulge from the LR/AR associated with BhCas12b did not impinge on GFP silencing (Fig. 2c), likely due to the bulge falling outside of the nuclease binding region[57]. Using the bulge-removed variant as a baseline to interrogate programmability of the LR/AR and SR/AR RNA stems (Fig. 2c), we found that both stems could be reprogrammed without impinging on GFP silencing, whether changing the lower or upper portion of the LR/AR stem (cr1-4) or the SR/AR stem (cr5-6).

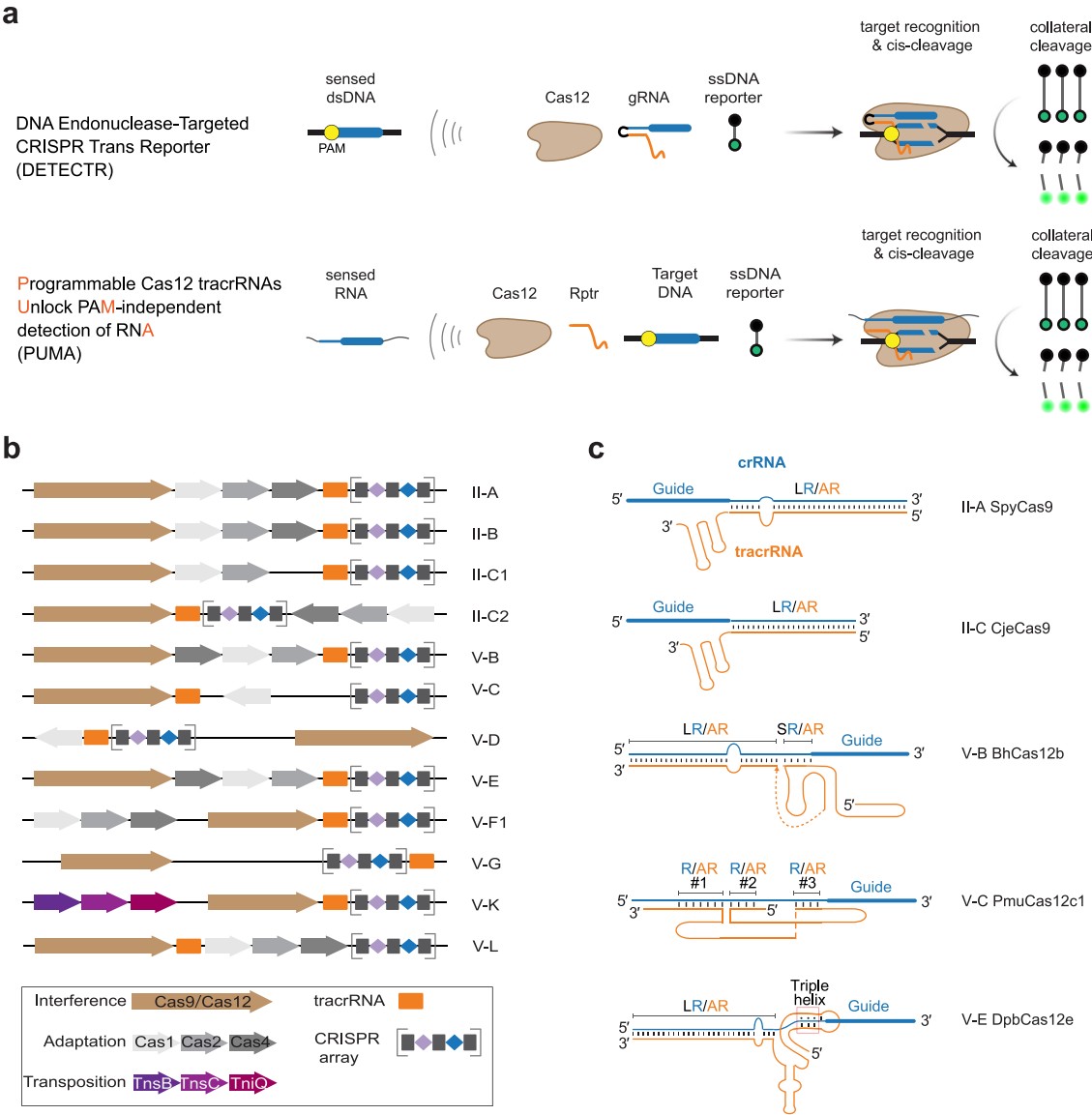

**Fig. 1 | A framework for PAM-independent RNA detection by tracrRNA-dependent Cas12 nucleases with PUMA. a** Overview of Cas12 Rptr-based RNA detection platform. Top: Traditional Cas12-based diagnostics use gRNAs to recognize and cut dsDNA targets, triggering trans-cleavage activity for signal visualization and amplification. However, selected targets must be flanked by a PAM sequence and cannot directly detect RNA. Bottom, Reprogrammed tracrRNAs (Rptrs), binding and converting sensed RNAs into guide RNAs, enable direct and PAM-independent RNA detection. **b** Known tracrRNA-encoding CRISPR-Cas systems. The tracrRNA is encoded in all Type II subtypes and in 8 of 14 defined Type V subtypes, including V-B, -C, -D, -E, -F, -G, -K, and -L. The CRISPR locus architecture is

based on prior work[5,6,11]. **c** Representative examples of five different duplexes formed between the tracrRNA and crRNA among tracrRNA-encoding CRISPR-Cas systems. In type II CRISPR-Cas systems, a single imperfect RNA duplex is formed between the crRNA repeat and tracrRNA anti-repeat (R/AR). In contrast, the V-B, -F, and -G systems form two duplexes (long repeat/anti-repeat duplex, LR/AR; short repeat/anti-repeat duplex, SR/AR), the V-E and -K systems form one LR/AR duplex and one triple helix, and the V-C system forms three R/AR duplexes. PmuCas12c1, Cas12c from *Parasutterella muris*, PDB: 7VYX[90]. See detailed information in Supplementary Fig. 1.

The crRNA-tracrRNA duplex could be similarly reprogrammed for the *Bacillus thermoamylovorans* Cas12b (BthCas12b)[57], as changes in the LR/AR and SR/AR of a fused single-guide RNA (sgRNA) were well tolerated (Supplementary Fig. 2). Therefore, tracrRNAs associated with Cas12b nucleases are highly amenable to reprogramming.

We next explored the extent to which the BhCas12b Rptr could be applied for RNA detection. We started with the *CJ8421_04975* mRNA previously used to evaluate Rptrs associated with different Cas9 nucleases[31]. Two Rptrs hybridizing at different loci of *CJ8421_04975* mRNA were designed based on rules derived from our mutational analysis of the LR/AR and SR/AR (Fig. 2c and Supplementary Fig. 3). Strong GFP silencing was observed for both BhRptr1

and BhRptr2 when compared with the non-targeting crRNA control, and both Rptrs combined with the mRNA exhibited similar performance as their equivalent crRNA/tracrRNA pairs (Fig. 2d, e). Furthermore, dsDNA targeting occurred specifically through the predicted guide sequence, as mutating the predicted seed region or scrambling the tracrRNA anti-repeat (long, short or both) fully inhibited GFP silencing. The one exception was scrambling the anti-repeat of the tracrRNA associated with locus 1 of the mRNA, which still maintained substantial targeting activity likely due to shifted base pairing in the short duplex (Fig. 2d, e). Overall, the Cas12b tracrRNA can be reprogrammed to link an RNA-of-interest to sequence-specific dsDNA targeting.

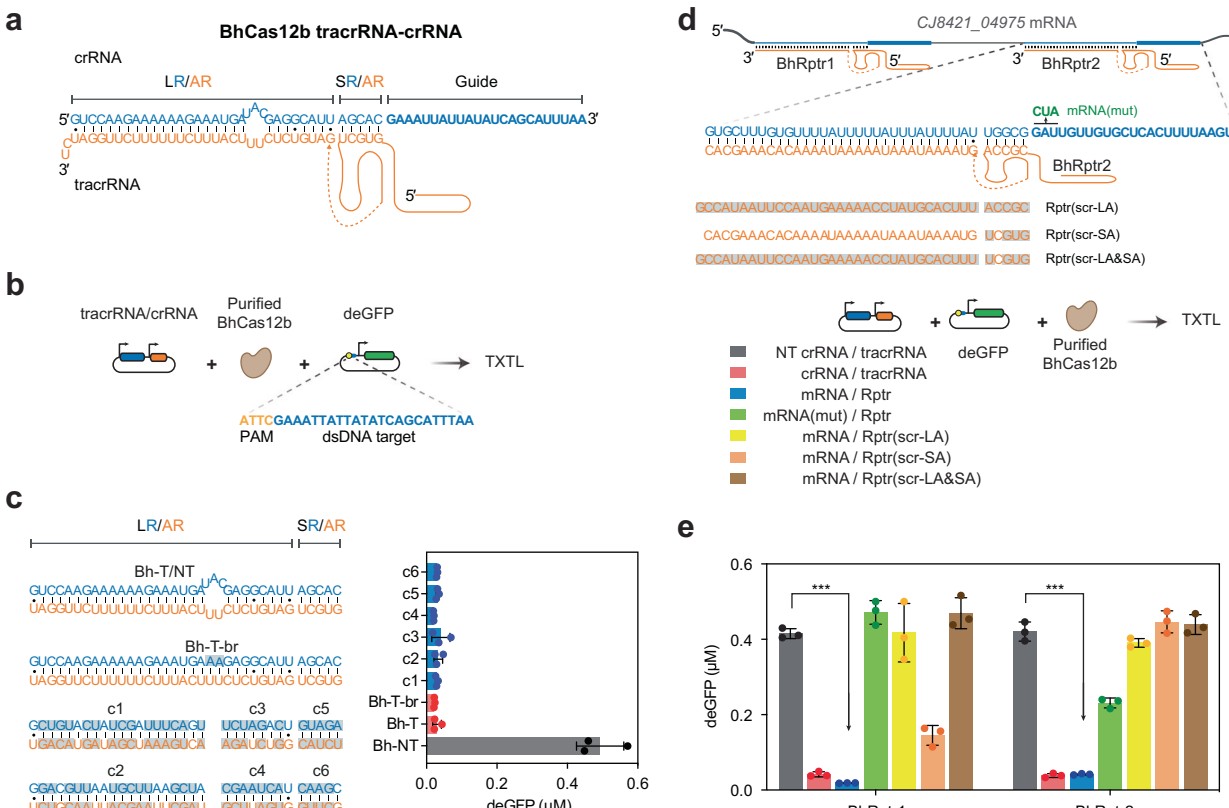

**Fig. 2 | TracrRNA reprogramming enables sequence-specific RNA detection by Cas12b. a** Predicted tracrRNA-crRNA structure for BhCas12b based on its ortholog BthCas12b (PDB: 5WTI[57]). R/AR, repeat/anti-repeat. **b** Setup to assess Rptr functionality using cell-free transcription-translation (TXTL). Expressed Cas-guide RNA complex recognizes and cuts its dsDNA target, causing the degradation of target-encoding GFP reporter plasmid and resulting in lower fluorescence compared to a non-targeting guide control. **c** 16-hour endpoint fluorescence measurements in TXTL when changing the long and short RNA duplexes. NT, non-targeting guide; T, targeting guide; T-br, targeting crRNA with bulge removed. **d** Setup to reprogram tracrRNAs to sense a *Campylobacter jejuni* transcript *CJ8421_04975* mRNA. The guide and target components are added in the form of DNA constructs, while the

purified BhCas12b protein is used. mRNA(mut), mRNA with point mutations in the predicted seed region of the guide. Rptr(scr-LA), Rptr with the long anti-repeat sequence scrambled; Rptr(scr-SA), Rptr with the short anti-repeat sequence scrambled; Rptr(scr-LA&SA), Rptr with both long and short anti-repeat sequence scrambled. **e** 16-hour endpoint fluorescence measurements in TXTL when assessing Rptr-guided sequence-specific dsDNA targeting. Nucleotide changes in R/AR stems in **c** and **d** are indicated by gray boxes. Bars and error bars in **c**, **e** represent the mean and standard deviation from three independently mixed TXTL reactions. Dots represent individual measurements. \*\*\*$p < 0.001$ based on a one-sided Student's t-test with unequal variance ($n = 3$). Source data are provided as a Source Data file.

## Reprogrammed tracrRNAs link RNA sensing and dsDNA targeting by Cas12e and Cas12f1 nucleases

Building on the reprogramming of Cas12b tracrRNAs, we turned to the *Acidibacillus sulfuroxidans* Cas12f1 (AsCas12f1) from Type V-F CRISPR-Cas systems[39,64]. While its crRNA-tracrRNA duplex parallels that associated with BhCas12b (Fig. 3a), AsCas12f1 is a much smaller protein and forms a homodimer when binding a single crRNA-tracrRNA duplex. Using the TXTL assay with plasmid-expressed AsCas12f1 and an sgRNA, we found that the intervening bulge was also dispensable and the LR/AR and SR/AR could be fully reprogrammed without impinging on GFP silencing (Fig. 3b). The base-pairing in the SR/AR was crucial for dsDNA targeting, as deletion of the SR portion of the SR/AR or mismatches in the SR/AR substantially inhibited GFP silencing (Fig. 3b). We further demonstrated that three Rptrs designed to hybridize to different loci in the *CJ8421_04975* mRNA yielded GFP silencing with comparable performance as their equivalent crRNA/tracrRNA counterparts in TXTL (Fig. 3c, d). As before, mutating the seed region in the predicted guide or scrambling the tracrRNA anti-repeat (long, short or both) fully inhibited GFP silencing.

Deviating from Cas12b and Cas12f1, Cas12e nucleases rely on crRNA-tracrRNA duplexes containing an RNA triple helix instead of a pseudoknot (Fig. 3e and Supplementary Fig. 1)[35,53,61,62], posing an even greater challenge for RNA detection with PUMA. We selected the

previously characterized Deltaproteobacteria Cas12e (DpbCas12e)[35] and evaluated the reprogrammability of the bulged stem as well as the triple helix. Paralleling BhCas12b and AsCas12f1, removing the bulge and a G-U wobble pair in the context of an sgRNA did not compromise GFP silencing, and the stem could be readily reprogrammed (Fig. 3f). Turning to the triple helix, this helix is formed by two separate tracts of three uracils at the 5′ end of the tracrRNA sandwiching three adenosines in the repeat (Fig. 3e and Supplementary Fig. 1)[35]. A *cis* Hoogsteen/Watson-Crick base pair forms between the U·A and a *cis* Watson-Crick/Watson-Crick base pair forms between the A-U, assigning the triple helix to the cWW/cHW triple family[65]. RNA triple-helix motifs are found in various functional RNAs, such as telomerase RNAs[66,67], riboswitches[68] and long noncoding RNAs[69,70]. Despite its diversified distribution, the changeability of RNA triple helix in these biologically important RNAs has not been systematically investigated.

We reasoned that other RNA triple helices in the same cWW/cHW family might preserve dsDNA targeting by DpbCas12e. Using the RNA Base Triple Database as a reference[71], we tested all nine RNA triple helix combinations reported in existing functional RNAs (s11-s18, and the native U.A-U), three expected to form a triple helix but not observed to-date (s19-21), and two not expected to form a triple helix and not observed to-date (s22-23, Fig. 3g). Among the 14 tested triple-helix combinations, two (C.G-U_s11, C.G-C_s18) yielded GFP silencing

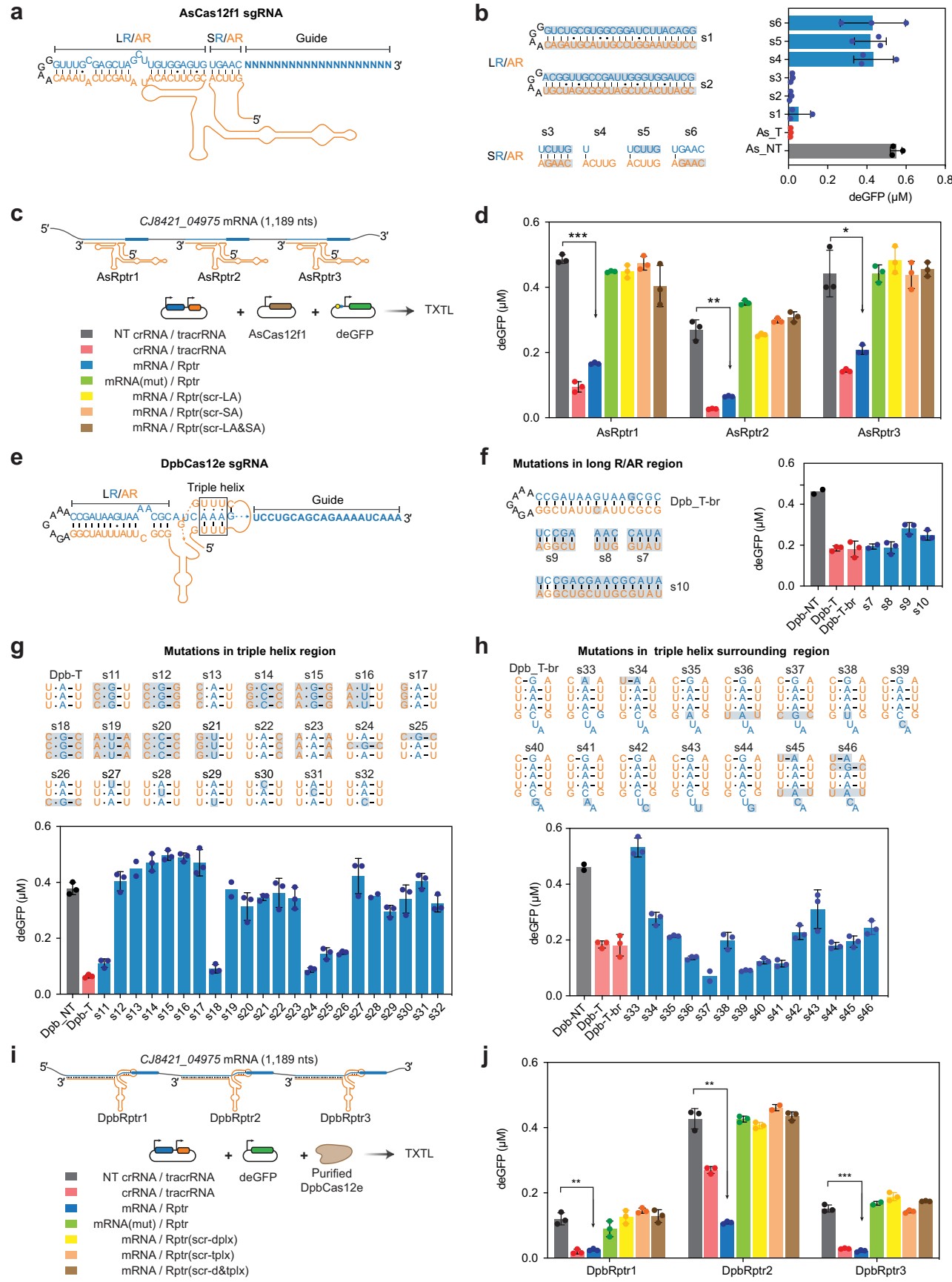

**Fig. 3 | TracrRNA reprogramming enables sequence-specific RNA detection by Cas12f1 and Cas12e. a** AsCas12f1 sgRNA structure (PDB: 8J12[64]). See the detailed information in Supplementary Fig. 1. **b** 16-hour endpoint fluorescence measurements in TXTL when reprogramming the long and short RNA duplexes in the AsCas12f1 sgRNA. NT, non-targeting crRNA; T, targeting crRNA. **c** Setup to detect the *Campylobacter jejuni* transcript *CJ8421_04975* mRNA using AsCas12f1 Rptrs in TXTL. **d** 16-hour endpoint fluorescence measurements in TXTL for Rptr-guided sequence-specific dsDNA targeting by AsCas12f1 in TXTL. **e** Structure of DpbCas12e sgRNA (PDB: 6NY3)[35]. In the triple-helix region, a cis Hoogsteen/Watson-Crick base pair is formed between the U.A and a cis Watson-Crick/Watson-Crick base pair between the A-U. **f** 16-hour endpoint fluorescence measurements in TXTL when assessing the changeability of the LR/AR region. Dpb_T-br, targeting sgRNA with the bulge and G.U wobble base pair removed. **g** 16-hour endpoint fluorescence measurements in TXTL when changing the RNA triple-helix region. **h** 16-

hour endpoint fluorescence measurements in TXTL when changing the RNA triple-helix surrounding region. **i**, Setup to detect the *Campylobacter jejuni CJ8421_04975* mRNA using DpbCas12e Rptrs in TXTL. **j**, 16-hour endpoint fluorescence measurements for Rptr-guided sequence-specific dsDNA targeting by DpbCas12e in TXTL. Rptr(scr-dplx), Rptr with a scrambled anti-repeat sequence; Rptr(scr-tplx), Rptr with the RNA triple-helix sequence scrambled; Rptr(scr-d&tplx), Rptr with the RNA duplex and triple-helix sequence scrambled. Nucleotide changes in AsCas12f1 sgRNA and DpbCas12e sgRNA in **b, f, g** and **h** are indicated by gray boxes. Bars and error bars in **b, d, f, g, h,** and **j** represent the mean and standard deviation from three independently mixed TXTL reactions. Dots represent individual measurements. No error bars are shown when only two replicates were successfully collected. *: $p < 0.05$. **: $p < 0.01$. ***: $p < 0.001$ based on a one-sided Student's t-test with unequal variance ($n = 3$). Source data are provided as a Source Data file.

comparable to that of the native U.A-U. In addition, installing the combination of U.A-U and C.G-C base triples in the RNA triple-helix region (s24-26) also yielded comparable GFP silencing. As expected, disrupting the RNA triple-helix conformation in one of the three triples in the triple-helix region abolished dsDNA targeting (s27-32, Fig. 3g), indicating a stringent triple-helix conformation required by DpbCas12e.

The RNA triple-helix region is surrounded by one C-G base pair at the 3′ end and three unpaired nucleotides (AUC) at the 5′ end of the repeat that may also represent necessary sequence or structural features (Fig. 3h). For the C-G base pair, we found that introducing a C.A mismatch (s33) fully abolished silencing, while changing the base pair to U-A (s34) only modestly reduced GFP silencing (Fig. 3h). For the AUC at the 5′ end, mutating the C to A, G and U (s35-s38) resulted in similar or even improved GFP silencing (Fig. 3h). The U could also be replaced with other nucleotides (s39-s41) without compromising activity (Fig. 3h). Changing the A to C or G (s42, s44) was also well tolerated, while changing the A to U (s43) substantially inhibited GFP silencing (Fig. 3h). Together, the RNA duplex and triple-helix regions are reprogrammable, albeit with less flexibility for the triple-helix region (Supplementary Fig. 2).

Based on the insights from the systematic mutational analyses to DpbCas12e sgRNA, we designed three Rptrs targeting different loci in *CJ8421_04975* mRNA (Fig. 3i). We observed substantial GFP silencing for all three designed Rptrs, with comparable performance to that of their equivalent crRNA:tracrRNA pairs. As before, mutating the seed region in the predicted guide or scrambling the tracrRNA anti-repeat inhibited GFP silencing (Fig. 3j). Overall, Rptrs could be extended to different Cas12 nucleases with varying tracrRNA-crRNA structures.

**Truncating the dsDNA target enhances collateral cleavage-based RNA detection with BhCas12b and DpbCas12e**

In contrast to Cas9, Cas12 non-specifically cleaves ssDNA upon target recognition, enabling signal amplification as part of CRISPR-based diagnostics[2]. We therefore reasoned that combining Rptrs, dsDNA targets, and ssDNA reporters would couple RNA detection by Cas12 to an amplified readable output–the basis of PUMA. To assess the collateral effects of BhCas12b, we devised an in vitro collateral cleavage assay using purified BhCas12b protein, in vitro-transcribed sgRNAs or sensed RNAs and Rptrs, linear dsDNA targets and a ssDNA fluorophore-quencher reporter (Fig. 4a). Upon recognition and cleavage of its dsDNA target, the nuclease non-specifically cleaves the fluorophore-quencher reporter, resulting in an increase in fluorescence.

We began with an sgRNA and a 334-bp linear dsDNA containing a 27-bp PAM-flanked target, with the resulting in vitro reaction conducted at 37 °C (Supplementary Fig. 4a). We observed slight background fluorescence without the dsDNA target and monotonically increasing fluorescence with the dsDNA target that plateaued after 12 hours ($k_{obs} = 0.03\,h^{-1}$, Supplementary Fig. 4b, Supplementary Data 1), in line with *cis*-cleavage of the dsDNA target triggering multi-

turnover collateral cleavage of the fluorescent ssDNA reporter by BhCas12b. The activity exhibited by BhCas12b was weaker compared to that by FnCas12a ($k_{obs} = 0.11\,h^{-1}$), DpbCas12e ($k_{obs} = 0.65\,h^{-1}$) and LbCas12a ($k_{obs} = 2.1\,h^{-1}$) under equivalent conditions (Supplementary Fig. 4b, Supplementary Data 1). Elevating the temperature from 29 to 42 °C increased the reaction rate by 1.7-fold ($k_{obs} = 0.19\,h^{-1}$ at 42 °C) (Supplementary Fig. 4c, Supplementary Data 1), in line with higher temperatures yielding optimal cleavage activity for Cas12 nucleases[9,36,52].

With an in vitro collateral cleavage assay in place, we next turned to the dsDNA target. Standard Cas12-based diagnostics have little control over the composition of the dsDNA target without extensive manipulations. In contrast, the dsDNA target is provided as part of PUMA, granting complete control over its sequence, length, and chemistry. This control in turn could be leveraged to enhance the reaction. As a start, we evaluated the impact of using targets encoded on shorter linear DNA, perceivably by reducing the search time for the target sequence. In line with this rationale, we observed a 5.6-fold increase in collateral cleavage activity at 37 °C when shortening the dsDNA target length from 334 bp ($k_{obs} = 0.03\,h^{-1}$) to 94 bp ($k_{obs} = 0.17\,h^{-1}$). However, collateral cleavage activity decreased when shortening the DNA length to 60 bp ($k_{obs} = 0.12\,h^{-1}$) or to 48 bp ($k_{obs} = 0.08\,h^{-1}$) (Supplementary Fig. 4d, Supplementary Data 1). We also tested ssDNA targets, which exhibited at least a 2-fold increase in collateral activity than dsDNA targets of equivalent size (Supplementary Fig. 5), in line with circumventing PAM recognition and DNA unwinding. We continued to use dsDNA targets though due to their more stringent and specific target recognition[2].

The observed impact of DNA length on signal production led us to explore a distinct aspect of the dsDNA target: the extent of cleavage by Cas12 nucleases. Upon target recognition, Cas12 nicks the non-target strand followed by the target strand of the dsDNA target through the nuclease's RuvC domain[72,73], leading to a cleaved dsDNA target with a 5′ overhang (Fig. 4a). Complete cleavage of the dsDNA target normally precedes collateral cleavage[72], with target strand cleavage posing the rate-limiting step[36,73-75]. We therefore hypothesized that using a dsDNA target with a processed target strand would increase the observed rate of collateral cleavage. In line with this hypothesis, a dsDNA target with a processed non-target strand yielded similar collateral cleavage rates to that of an unprocessed dsDNA target ($k_{obs} = 0.06 - 0.16\,h^{-1}$) at 37 °C for dsDNA lengths ranging between 45 and 94 bp (Fig. 4b, Supplementary Fig. 6a, b, and Supplementary Data 1). In contrast, a dsDNA target with a processed target strand yielded increased collateral cleavage rates ($k_{obs}$ up to $0.46\,h^{-1}$ for NTS + 55: TS + 0), in line with target strand cleavage posing the rate-limiting step (Fig. 4b and Supplementary Fig. 6a, b). A similar collateral cleavage rate ($k_{obs} = 0.44\,h^{-1}$) was observed for a dsDNA target with both strands processed (NTS-6: TS + 0) (Supplementary Fig. 6a, b). Finally, trimming the target strand by two additional nts towards the PAM can further enhance the observed collateral cleavage activity (NTS-6: TS-2, $k_{obs} = 0.56\,h^{-1}$)

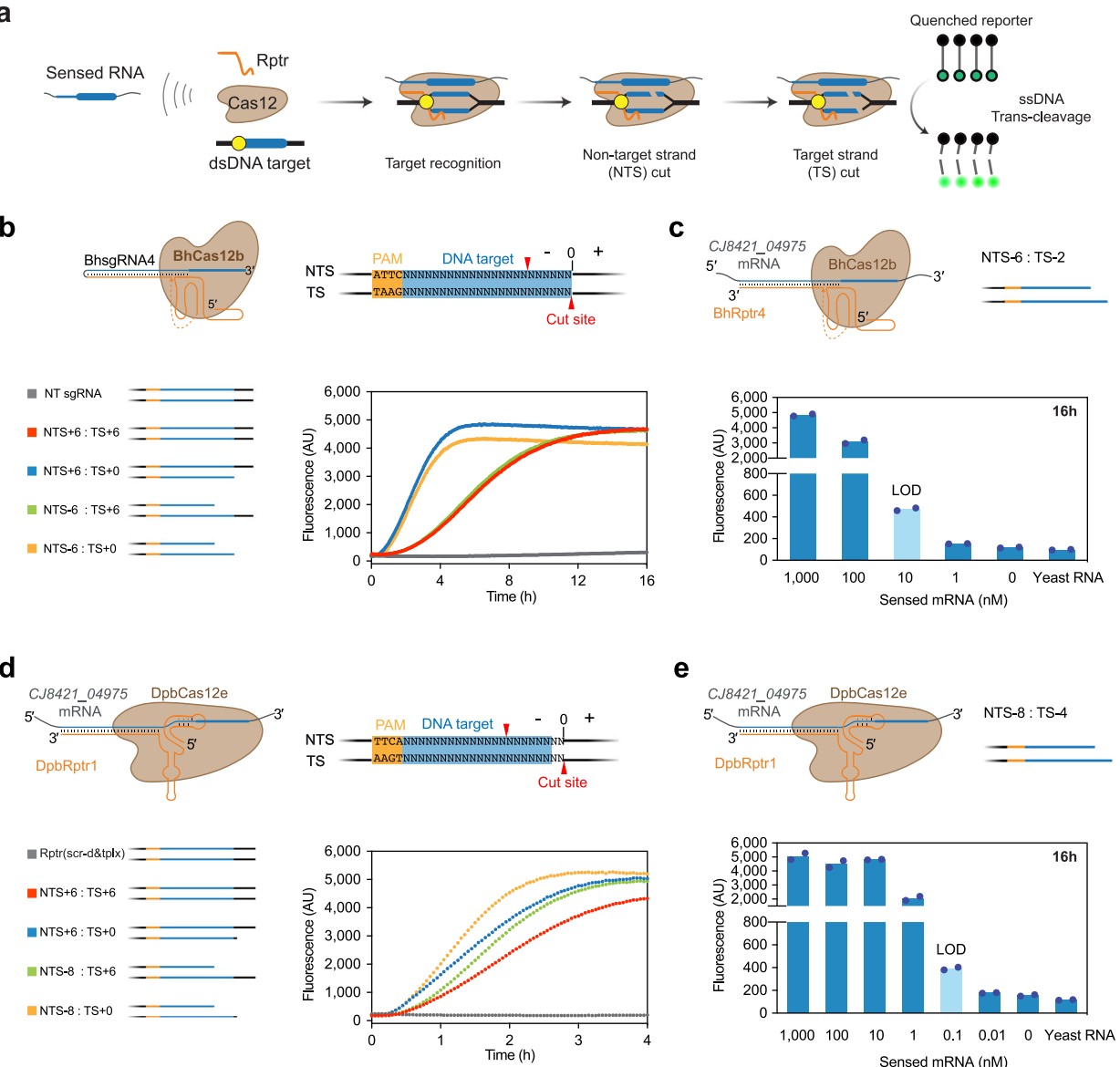

**Fig. 4 | Truncating dsDNA targets enhances collateral cleavage activity.**
**a** Schematic of the in vitro trans-cleavage assay. The assay includes purified a Cas12 nuclease, an in vitro transcribed Rptr, and a linear dsDNA target. The Cas12-guide RNA ribonucleoprotein (RNP) recognizes and cleaves its dsDNA target, which triggers non-specific cleavage activity on ssDNA. Specifically, cleavage of the non-target strand (NTS) occurs before cleavage of the target strand (TS). F, fluorophore; Q, quencher. Yellow circle, PAM; **b** Impact of unprocessed or processed targets on in vitro trans-cleavage activity by BhCas12b. TS cleavage is the rate-limiting step. Red arrow, cleavage site. The cleavage site of TS is set as position 0. -, truncating the target sequence on NTS or TS. +, adding an overhang on NTS or TS. The PAM is in brown and the target is in blue. **c**, Direct detection of the full-length *CJ8421_04975* mRNA by BhCas12b based on in vitro collateral cleavage activity. Yeast RNA is added in the same mass amount as the 1000 nM sensed mRNA, and

the best-performing dsDNA target NTS-6: TS-2 is used. **d**, Impact of unprocessed or processed targets on in vitro collateral cleavage activity by DpbCas12e. **e** Direct detection of the full-length *CJ8421_04975* mRNA by DpbCas12e based on in vitro collateral cleavage activity. Yeast RNA is added in the same mass amount as the 1000 nM sensed mRNA, and the best-performed dsDNA target NTS-8: TS-4 is used. 16 h end-point values were used to make the plots in **c** and **e**. See Supplementary Figs. 6b and 10a for the complete time courses. Curves in **b** and **d** represent the mean from two independent collateral assays. Bars and dots in **c** and **e** represent the mean and individual measurements, respectively, from two independent collateral cleavage assays. Light blue bars indicate the limit-of-detection (LOD) conservatively estimated as the lowest concentration yielding an average fluorescence exceeding 50% of that of the no-RNA control. Source data are provided as a Source Data file.

(Supplementary Fig. 6a, b). With conditions established for enhanced RNA detection using BhCas12b, we turned to detecting the full-length *CJ8421_04975* mRNA using BhRptr4 in vitro. Under the optimal conditions with the shortest and processed dsDNA target (NTS-6:TS-2) at 42 °C, the sensed mRNA was detected at 1 µM in 45 minutes and at 10 nM in 16 hours based on endpoint measurements compared to a no-RNA control (Fig. 4c and Supplementary Fig. 6c).

The optimized experimental setup with BhCas12b allowed us to assess how the ability of the sensed RNA and Rptr to hybridize impacts

collateral cleavage activity. One potential factor is the formation of internal secondary structures that hinder hybridization. To test this factor directly, we introduced extensions to the 5′ extensions to the ncrRNA associated with BhRptr4 (Supplementary Fig. 7). The hairpins reduced collateral cleavage activity, with an internal hairpin inhibiting more strongly than a flanking hairpin. In the absence of these structures, introducing an annealing step did not enhance collateral cleavage activity (Supplementary Fig. 8). Of note, collateral cleavage activity resulting from pairing of the partial *CJ8421_04975* mRNA

fragment and BhRptr4 was higher than that obtained with the equivalent sgRNA, indicating that hybridization between a sensed RNA and Rptr is not necessarily a bottleneck to RNA detection.

With factors influencing RNA detection with BhCas12b established, we asked whether increasing the reaction temperature and truncating the dsDNA target also apply to DpbCas12e, which exhibited much higher collateral cleavage activities (Supplementary Fig. 4). We tested DpbCas12e with DpbRptr1 against the full-length *CJ8421_04975* mRNA along with different-sized dsDNA targets at different temperatures (Supplementary Fig. 9a–c). Similar to BhCas12b, DpbCas12e exhibited increased activity when elevating the temperature ($k_{obs} = 0.07\,h^{-1}$ at 29 °C, $0.43\,h^{-1}$ at 37 °C and $0.67\,h^{-1}$ at 42 °C) (Supplementary Fig. 9b) and when shortening the length of the dsDNA target ($k_{obs} = 0.43\,h^{-1}$ for 331 bp and $0.54\,h^{-1}$ for 44 bp) (Supplementary Fig. 9c). Moreover, introducing a processed dsDNA target increased the collateral cleavage rate (for 91-bp target, $k_{obs} = 0.46\,h^{-1}$ for unprocessed strands, $0.78\,h^{-1}$ for processed target strand) (Fig. 4d and Supplementary Fig. 10a). Finally, under the optimized conditions using the double-strand processed 38-bp dsDNA target at 42 °C, the sensed mRNA could be detected at a concentration of 1 μM in 9 minutes and at a concentration of 0.1 nM in 16 hours (Fig. 4e and Supplementary Fig. 10b). Therefore, different tracrRNA-dependent Cas12 nucleases can be co-opted for direct, PAM-independent RNA detection in vitro.

## Reprogrammed tracrRNAs circumvent background collateral activity from Cas12-gRNA complexes

When comparing collateral cleavage activities across Cas12 orthologs (Supplementary Fig. 4b), we noticed that the LbCas12a-gRNA complex produced substantial fluorescence even in the absence of its corresponding dsDNA target, reaching approximately 70% of the levels seen when its dsDNA target is present after 16 hours of incubation (Supplementary Fig. 4b). A high background activity was also reported for AsCas12a in previous studies[76,77]. To assess the prevalence of this background activity, we tested four BhCas12b sgRNAs (#1-4, with the #4 guide used with other Cas12 orthologs in Figure S4b) using processed dsDNA targets (NTS-6: TS-2). Substantial DNA target-independent activity was observed for sgRNA#1 and #2, with comparable fluorescence levels to those with the dsDNA targets after 16 hours (Fig. 5a). Intriguingly, sgRNA#1 exhibited high cleavage activity ($k_{obs} = 1.02\text{-}1.16\,h^{-1}$) regardless of the presence or absence of the dsDNA target. This phenomenon was not isolated, as 5 out of 10 additional sgRNAs we tested exhibited DNA target-independent collateral activity higher than that of sgRNA#4 (Supplementary Fig. 11).

This DNA target-independent collateral activity would reduce the sensitivity of nucleic-acid detection, making it more challenging to identify low-concentration biomarkers. In contrast, we hypothesized that any background activities would be greatly reduced using Rptrs, as the guide RNA is principally formed only in the presence of the sensed RNA. Supporting this hypothesis, combining a sensed RNA and Rptr for BhCas12b drove collateral activity even in the absence of the DNA target (Supplementary Fig. 12). In the absence of the sensed RNA, each Rptr alone resulted in endpoint fluorescence levels 3.5-fold to 29.9-fold lower than those observed in the presence of the corresponding sgRNA (Fig. 5b). Based on this difference, we directly compared the sensitivity of BhCas12b detecting dsDNA with an sgRNA or detecting the equivalent RNA with a Rptr. The limit-of-detection was around 10-fold lower using a Rptr than an sgRNA for one site (#4), while RNA detection (with a Rptr) but not DNA detection (with an sgRNA) was possible at another site (#1) (Fig. 5c). Thus, the sensitivity of nucleic-acid detection with Cas12 nucleases can be enhanced by detecting RNA with Rptrs rather than detecting DNA with sgRNAs, at least depending on the nuclease and detected sequence.

Given the enhanced sensitivity when detecting RNA with PUMA versus dsDNA traditionally detected with Cas12 nucleases, we asked how PUMA compares to the two standard CRISPR-based diagnostic approaches DETECTR for DNA detection with Cas12[2] and SHERLOCK for RNA detection with Cas13[4] (Supplementary Fig. 13). We chose to detect three loci within the *CJ8421_04975* DNA/mRNA and used sensitivity as the basis of comparison. BhCas12b was used for both PUMA and DETECTR to ensure a direct comparison, while PbuCas13b was used for SHERLOCK[78]. No pre-amplification was included to directly gauge the sensitivity associated with each Cas nuclease. Two of the sites lacked the PAM recognized by BhCas12b, in line with the requirement for a PAM inherent to DETECTR. Of the detected loci, the three approaches performed similarly, with the measured limit of detection either at 1 nM or 10 nM. Thus, PUMA can perform similarly to DETECTR and SHERLOCK, at least with the tested Cas nucleases, with PUMA targeting a broader range of sites than DETECTR.

## A universal Cas12 Rptr can decipher different bacterial pathogens

One core feature of Rptrs is that base pairing with a sensed RNA is somewhat flexible, whereas the flanking guide sequence should direct dsDNA targeting that is highly sensitive to mismatches[31]. To exploit this feature, we applied Cas12 Rptrs to differentiate bacterial pathogens based on their 16 S rRNA[79,80]. Differentiating pathogens can be important to select appropriate courses of treatment for different indications such as acute sepsis, urinary tract infections, or sexually transmitted diseases. Traditional CRISPR diagnostics based on collateral cleavage by Cas12 or Cas13-based diagnostics have taken strides in this direction[81], with one example using multiple guide RNAs to detect different bacterial pathogens[82]. In contrast, with this core feature of Rptrs, a single Rtpr could be designed to pair next to a variable region of 16 S rRNA indicating the genus. The variable region would then be matched to a dsDNA target, with its cleavage and subsequent collateral activity indicating which pathogen is present.

To determine how to best design the Rptr, we began by evaluating the specificity of the three different Cas12b homologs (BthCas12b, BhCas12b and AacCas12b), with the goal of identifying at least one homolog exhibiting high guide-target mismatch sensitivity. We assessed collateral cleavage activity of each homolog using a Rptr-sensed RNA encoding the same guide sequence along with dsDNA targets containing two consecutive mismatches sliding through the guide-target region (Fig. 6a). Among the three orthologs, BthCas12b was the most sensitive to guide-target mismatches, especially in positions 5-12 proximal to the PAM in which the mismatches reduced collateral cleavage activity between $10^2$-fold and $10^5$-fold (Fig. 6a). We also evaluated the extent to which BthCas12b accepts mismatches between the sensed RNA and the Rptr (Supplementary Fig. 14). Mismatches in the long or short repeat consistently reduced but rarely eliminated activation of collateral cleavage activity even with four consecutive mismatches. This flexibility lends to pairing with conserved 16 S rRNA regions with some variability, even if unintended RNA duplexes bound by Cas12 could be generated in the process. We therefore proceeded with BthCas12b and aimed for sequence differences to fall within the most sensitive positions of the target.

Following this approach, we designed a single BthCas12b Rptr that hybridizes to a conserved region of bacterial 16 S rRNA, with the downstream variable region serving as the guide sequence. We specifically focused on five common bacterial pathogens, *E. coli*, *Klebsiella pneumoniae*, *Staphylococcus aureus*, *Enterococcus faecalis*, and *Listeria monocytogenes* (Fig. 6b), where the sequence differences fall within the region of mismatch sensitivity for BthCas12b. As before, a PAM did not need to appear within the sensed RNA, as this was encoded within the dsDNA targets. We then assessed collateral cleavage activity for each 16 S rRNA fragment and each dsDNA target. The fragment was introduced at a final concentration of 100 nM, reflecting the output of isothermal pre-amplification and in vitro transcription[2]. The presence of the 16 S rRNA fragment from one specific pathogen triggers fluorescence release only when paired with its corresponding dsDNA target

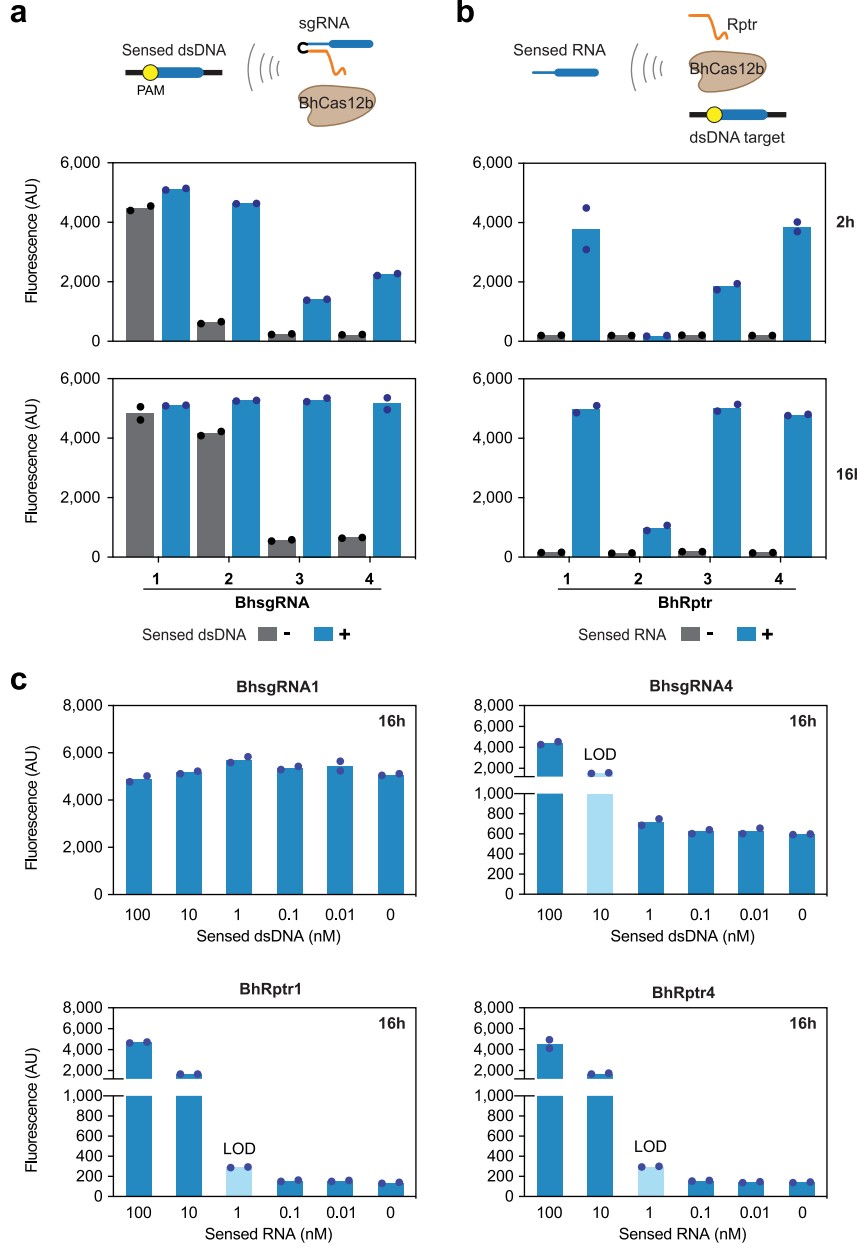

**Fig. 5 | PUMA exhibits reduced background collateral activity due to the absence of a traditional single-guide RNA. a**, Measured in vitro collateral cleavage activity with BhCas12b and an sgRNA with or without a dsDNA target. **b**, Measured in vitro collateral cleavage activity with BhCas12b and a Rptr and a dsDNA target with or without the sensed RNA. **c**, Sensitivity comparison between sgRNA and Rptr. In **a-b**, 37-bp NTS-2:TS-2 processed dsDNA targets were used for both sgRNA and Rptr. In **c**, 334-bp DNA fragments containing the core PAM-flanking target were used with the sgRNAs and 37-bp NTS-2:TS-2 processed dsDNA targets were used with the Rptrs. In **a-c**, sgRNA#1 and sgRNA#4 share the same guide sequences as those generated by Rptr#1 and Rptr#4, respectively. Dots represent individual measurements from two independent collateral cleavage assays. Bars represent the mean of the dots. In **a-b**, values represent fluorescence measurements after reaction times of 2 hours and 16 hours. In **c**, values represent fluorescence measurements after reaction times of 16 hours. Light blue bars indicate the limit-of-detection (LOD) conservatively estimated as the lowest concentration yielding an average fluorescence exceeding 50% of that of the no-RNA control. Source data are provided as a Source Data file.

(Fig. 6c and Supplementary Fig. 15). We noticed 16 S rRNA from *L. monocytogenes* also gave rise to substantial fluorescence when pairing with the dsDNA target from *S. aureus*, likely due to the high similarity between their 16 S rRNA fragments with only three mismatches present outside of the seed region (Fig. 6b, c and Supplementary Fig. 15). Thus, specific detection of different pathogens based on 16 S rRNA can be achieved via a single Rptr.

## Discussion

Here we report the reprogramming of tracrRNAs associated with diverse type V CRISPR-Cas systems to address two general challenges

with traditional Cas12-based detection technologies: direct RNA detection and PAM requirements. Rptrs allow dsDNA-targeting Cas12 nucleases to directly detect RNA by binding and converting sensed RNAs into guide RNAs for dsDNA targeting. The sensed substrates are RNA, and the corresponding dsDNA targets can be supplied by users, who thus gain complete control over the sequence, length, and chemical properties of the dsDNA targets. The PAM is encoded into the dsDNA target, circumventing PAM requirements in the sensed RNA. We call the resulting approach for Cas12-based RNA detection via collateral activity PUMA. Virtually any RNA sequence can be detected with PUMA generally as long as the RNA sequence follows the

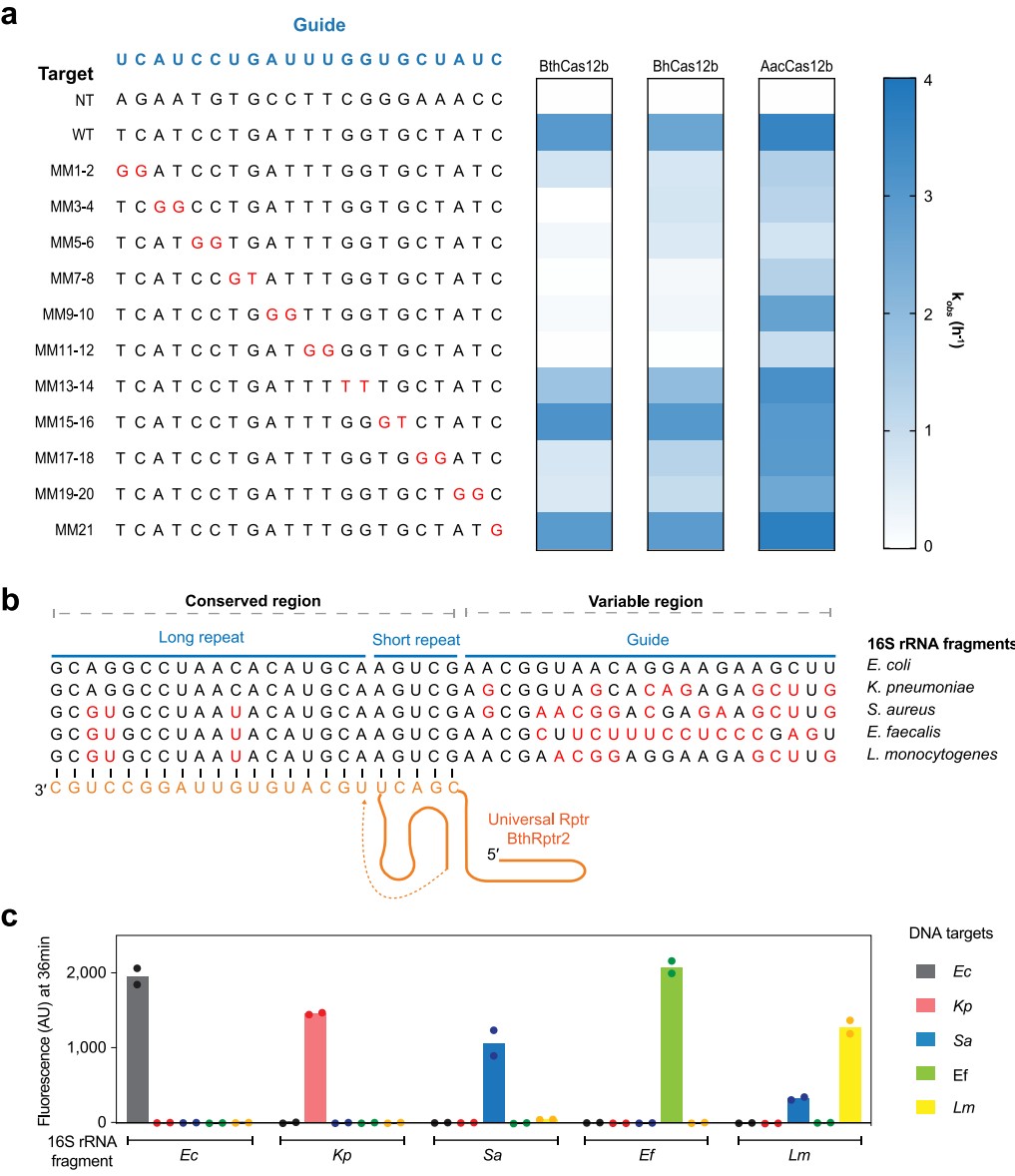

**Fig. 6 | A universal Rptr enables sequence-specific detection of 16 S rRNA from different bacterial pathogens. a** Tolerance of guide-target mismatches for three different Cas12b orthologs based on in vitro collateral cleavage activity. The DNA target is the same as the one used in Fig. 4B (BhsgRNA4 DNA target). Heat maps represent the mean kobs values from two independent collateral assays. See the $k_{obs}$ values in Supplementary Data 1. **b** Setup to differentiate 16 S rRNA from five different pathogens using only one universal BthCas12b Rptr binding to a conserved region of 16 S rRNA. A truncated long anti-repeat of 18 nts instead of the usual 31 nts is used in the universal Rptr. In the alignment, sequences that match the *E. coli* 16 S rRNA are in black, while those that do not match are shown in red. **c** Detection of pathogen 16 S rRNAs with a universal Rptr and corresponding dsDNA targets based on in vitro collateral cleavage activity with BthCas12b. Partial 16 S rRNA fragments of different pathogens at a final concentration of 100 nM were used. Values represent 36-minute reaction times. Values in **c** represent the mean and standard deviation from two independent collateral assays. Source data are provided as a Source Data file.

elucidated design rules (Supplementary Fig. 3) and meets a minimum length comprising an anti-repeat and guide. While interrogating PUMA, we found that shortened and processed dsDNA targets can boost the trans-cleavage activity and increase detection speed. We envision that this truncated dsDNA target strategy could also apply to other Cas12-based diagnostics, with the target strand in the dsDNA targets being processed by Type IIS restriction enzymes. Moreover, introducing synthetic mutations into the dsDNA targets could allow for RNA detection at a single-nucleotide level as applied previously[12,31]. Finally, while we relied on a fluorescence-based readout, collateral cleavage by Cas12 nucleases can also be linked to other readouts, such as lateral flow assays[83] or even smart materials[84].

One advantageous property of PUMA is the consistently low background collateral activity compared to traditional Cas12-based

detection technologies. For these traditional technologies, the background activities varied substantially among different sgRNAs used by Cas12b, indicating that the guide sequence could be a key determining factor. We speculate that the sgRNA induces conformational changes in the nuclease, exposing the RuvC active site and activating trans-cleavage activity even in the absence of a DNA target. However, the activation could be related to spurious byproducts of T7 transcription used to generate the sensed RNA and commonly used to generate sgRNAs for in vitro use. Further structural and functional studies of BhCas12b and other Cas12 nucleases as well as testing chemically-synthesized sgRNAs could reveal the underlying mechanism of this dsDNA target-independent activity, providing insights for guide RNA design and reduce or even eliminate this background activity. We recognize that background collateral activity could emerge from

promiscuous hybridization of the Rptrs to other RNAs in a sample, even if this possibility requires further exploration.

While we provided proof-of-principle demonstrations of PUMA, the approach still holds opportunities for further improvement. For one, nucleic-acid detection was associated with lower sensitivities (100 pM and 1 nM detection limits for DpbCas12e and BhCas12b, respectively), although we obtained similar sensitivities when testing DNA detection with BhCas12b and RNA detection with PbuCas13b akin to DETECTR[2] and SHERLOCK[4] without pre-amplification (Supplementary Fig. 13). Efforts could be devoted to screening other tracrRNA-dependent Cas12 orthologs for higher activities, or more advanced setups involving microfluidics could be employed to concentrate the detection components[85]. We also observed varying activities for Rptrs. As the folding pathways needed to form the pseudoknots and triple helix structures associated with the RNA duplexes are likely complex, folding predictions are unlikely to be sufficient to predict Rptr performance. Therefore, further work could elucidate design rules for highly effective Rptrs through machine learning based on high-throughput library screens.

Our previous LEOPARD platform achieved multiplex RNA detection in a single reaction by monitoring the sequence-specific cleavage of parallel DNA targets by Cas9 in a gel-based readout[31]. In the future, integrating the unique trans-cleavage activity of Cas12 Rptrs could enable sensitive RNA detection that exceeds that associated with LEOPARD. To do so, any combination would have to account for Cas12's non-specific trans-cleavage activity being inherently incompatible with multiplexed detection. One potential solution is the immobilization of DNA targets and reporters that would localize the collateral activities to a discrete location, such as an individual spot on a microarray or encapsulated within a water droplet. Therefore, the development of Cas12 Rptrs creates the opportunity to combine LEOPARD and PUMA for highly multiplexed and sensitive RNA detection.

We envision that Rptrs could also be extended to other tracrRNA-encoding Type V systems to achieve alternative functions beyond RNA detection. Cas12c possesses the unique capability of tracrRNA-assisted pre-crRNA processing within the guide region. This intriguing feature could be repurposed for precise RNA targeting, but with no trans-cleavage activity as observed for Cas12c2[38,49,50]. Precise RNA targeting could be achieved by using Rptrs to trigger the cis-cleavage in the resulting guide region in the target RNA. Furthermore, Rptrs could also be expanded to the Cas12k-guided transposition system, enabling transcription-dependent tissue-specific DNA integration[37,62,86]. By using a tissue-specific mRNA bound with Rptr as the source for guide RNA, the risk of off-targeting in undesired tissues or cell types could be minimized. These examples along with PAM-independent RNA detection with PUMA underscore the many technologies that could emerge from reprogramming tracrRNAs associated with Cas12 nucleases.

## Methods

### Strains, oligonucleotides, and plasmids
*E. coli* strain TOP10 was used for plasmid cloning. All primers, gBlocks used in this work were ordered from Integrated DNA Technologies. NEBuilder HiFi DNA Assembly Master Mix (New England Biolabs, cat. #E2621) was used for plasmid construction by Gibson assembly. Q5 site-directed mutagenesis kit (New England Biolabs, cat., #E0554S) was used for small insertion and nucleotide substitution. Unless otherwise specified, all nucleases used were expressed in plasmids with Cm selective marker and p15A origin-of-replication. All sgRNA, tracrRNA-crRNA, or Rptr-mRNA plasmids were expressed in plasmids with Amp selective marker and pUC origin-of-replication, and all targeted plasmids were expressed in plasmids with Kan selective marker and pSC101 origin-of-replication. *Bacillus thermoamylovorans* Cas12b (BthCas12b), *Alicyclobacillus acidoterrestris* Cas12b (AacCas12b), *Bacillus hisashii* Cas12b (BhCas12b), *Acidibacillus sulfuroxidans* Cas12f1

(AsCas12f1) and *Deltaproteobacteria* Cas12e (DpbCas12e) were PCR-amplified from plasmids pYPQ291 (Addgene ID #129671[87]), pYPQ290 (Addgene ID #129670[87]), pZ149-pcDNA3-BhCas12b_v4 (Addgene ID #122446[36]), pET28a-SUMO-AsCas12f (Addgene ID #171612[39]) and pCasX1 (Addgene ID #87685[88]), respectively, and subcloned to a PCR-linearized version of the vector pCB583 for TXTL and plasmid clearance assay, and subcloned to a PCR-linearized version of the vector pETM11(pBR322 ori) for protein expression and purification. The consensus PAM (5'-ATTN-3'[36]) was used for BhCas12b, BthCas12b and AacCas12b. The consensus PAMs (5'-NTTR-3'[39] and 5'-TTCN-3'[35,88], R = A, G) were used for AsCas12f1 and DpbCas12e, respectively. To construct targeted plasmids for the TXTL assay and plasmid clearance assay, target sequences were inserted directly upstream of the constitutive OR2-OR1 promoter with the template of pCB705 using Q5 site-directed mutagenesis. See Supplementary Data 2 for detailed information and links to annotated sequence maps in Benchling. Antibiotics were added where appropriate at final concentrations of 100 μg/ml for ampicillin, 34 μg/ml for chloramphenicol, and 50 μg/ml for kanamycin for *E. coli*.

### Protein expression and purification
The expression and purification of BhCas12b, BthCas12b, AacCas12b, and DpbCas12e were conducted through the recombinant protein expression facility of the Rudolf-Virchow-Center in Würzburg. These four Cas12 nucleases were cloned into a pETM11 expression plasmid (T7-6×His-SUMO-nuclease), respectively. Protein purification for these nucleases was conducted as previously described with modifications[35,36]. The resulting plasmids were expressed in the BL21 Rosetta *E. coli* strain. Cultures were grown to an OD of ~0.6, then expression was induced overnight at 18 °C by adding IPTG at a final concentration of 0.5 mM. Cells were harvested and frozen in liquid nitrogen. The cell pellets were resuspended in a lysis buffer of 500 mM NaCl, 10% glycerol, 1 mM TCEP, 20 mM imidazole and 20 mM Tris-HCl, pH 8.0 supplemented with protease inhibitors. Cells were lysed by sonication and pelleted at 4 °C and 35,000×g for 30 min. SUMO tag was removed by overnight digest at 4 °C with SenP2 SUMO-protease at a 1:50 weight ratio of protease to nuclease in a dialysis/ion exchange chromatography buffer of 300 mM NaCl, 10% glycerol, 1 mM DTT and 20 mM HEPES, pH 7.5. The HiTrap Heparin HP column was used to elute the cleaved proteins by gradually increasing NaCl from 0.3 M to 1 M. DpbCas12e eluted in two peaks from the HiTrap Heparin HP column, only the peak containing active protein was pooled as described previously[35]. Fractions containing BhCas12b, BthCas12b, AacCas12b and DpbCas12e were pooled, concentrated and loaded onto a Superdex 200 Increase column (GE Healthcare Life Sciences) with a final storage buffer of 300 mM NaCl, 10% glycerol, 2 mM DTT, 2 mM MgCl₂, and 20 mM HEPES, pH 8. Purified proteins were concentrated to 62.2 μM (BhCas12b), 80.5 μM (BthCas12b), 19.1 μM (AacCas12b), and 35 μM (DpbCas12e) stocks and flash-frozen in liquid nitrogen before storage at −80 °C. Purified PbuCas13b was provided by Wulf Blankenfeldt's group. All protein stocks were diluted to 1 μM with 1×Diluent B (300 mM NaCl, 10 mM Tris-HCl, 1 mM DTT, 0.1 mM EDTA, 500 μg/ml BSA, 50% Glycerol, pH 7.4, 25 °C). The diluted proteins were stored at −20 °C and used for enzymatic cleavage assays.

### Design of Rptrs and dsDNA targets
The *Campylobacter jejuni CJ8421_04975* mRNA was used as the primary template for sensed RNAs when evaluating Rptrs. Regions conforming to the design scheme for sensed RNAs in Supplementary Fig. 3 were chosen as targets for Rptrs. Repeat regions (R) in the sensed RNA bound by the anti-repeat region of Rptr (AR). The total length of the Rptr target was 31 nts for BhCas12b/BthCas12b/AacCas12b, 28 nts for AsCas12f1, and 37 nts for DpbCas12e. For BhCas12b, BthCas12b, Aac-Cas12b and AsCas12f1, the long and short anti-repeat regions of the native tracrRNA were replaced with the sequences complementary to

the chosen RNA targets. For DpbCas12e, the long anti-repeat region of the native tracrRNA was replaced with the sequences complementary to the chosen RNA target. Three RNA triple combinations U.A-U, C.G-C and U.G-C could be used in the RNA triple-helix region. For 16 S rRNA detection of multiple pathogens, one universal Rptr was designed to base pair with the conserved region of 16 S rRNA fragments from 5 different pathogens (*E. coli*, *Klebsiella pneumoniae*, *Staphylococcus aureus*, *Enterococcus faecalis* and *Listeria monocytogenes*). The 20-23 nt sequences upstream of these Rptr binding regions were treated as guides when designing dsDNA targets. The specific target length parallels that used for guides within sgRNAs for each Cas12[35,36,39]. For optimal performance in the collateral assay, it is recommended to use a minimal 5′ PAM sequence, with additional sequence extending at least 12 bp upstream of the target. Additionally, it is recommended to use a dsDNA target with both the non-target and target strands processed at the PAM-downstream end (See Fig. 4b, d as well as Supplementary Figs. 6b and 10a). See Supplementary Fig. 3 for illustrated design rules. ssDNA targets, with higher activity than dsDNA, could be utilized in the collateral assay despite potential challenges in distinguishing highly similar RNA sequences.

## Cell-free transcription-translation (TXTL) reactions

TXTL reactions were conducted as previously described with minor modifications[63]. The GFP variant deGFP was used as the fluorescent reporter for all TXTL assays. This variant was slightly modified for enhanced expression in TXTL[89]. All DNA fragments or plasmids used in the assay were free of nucleases (DNases, RNases) and inhibitors of the TXTL machinery (e.g., EDTA, ethidium bromide, SDS, Cl- ions). BthCas12b, sgRNA variants and deGFP-encoding reporter plasmids were added into myTXTL Sigma 70 Master Mix (Arbor Biosciences, cat. #507025) at a final concentration of 1 nM, 1 nM and 0.5 nM, respectively. AsCas12f1, sgRNA/tracrRNA-crRNA/Rptr-mRNA, and deGFP-encoding reporter plasmids were added into myTXTL Sigma 70 Master Mix at a final concentration of 3 nM, 0.5 nM and 0.1 nM, respectively. Purified protein BhCas12b/DpbCas12e, sgRNA/ tracrRNA-crRNA/Rptr-mRNA, and deGFP-encoding reporter plasmids were added into myTXTL Sigma 70 Master Mix at a final concentration of 100 nM, 0.5 nM and 0.1 nM, respectively. The DNA cleavage assay was performed in 5-μl reactions by measuring fluorescence on a Synergy Neo2 plate reader (BioTek) using a 96-well V-bottom plate (Corning Costar, cat. #3357) with an excitation filter of 485 nm and an emission filter of 528 nm. Time-courses were run for 16 h at 29 °C for AsCas12f1, 37 °C for BhCas12b, BthCas12b and DpbCas12e with an interval of 3 min between reads. The end-point values were used to make plots. A standard calibration curve produced with recombinant eGFP (Cell Biolabs, cat. #STA-201) was used to convert relative fluorescence units to the molar concentration of deGFP in the TXTL reaction.

## In vitro collateral cleavage assay

Sensed RNAs, Rptrs and crRNAs were encoded in double-stranded DNA fragments beginning with a T7 promoter and were transcribed using the HiScribe T7 in vitro transcription kit (New England Biolabs, cat. #E2040S). The transcribed RNAs were purified using RNA Clean & Concentrator-25 (Zymo Research) with residual DNA removed by In-Column DNase I treatment. The dsDNA targets were made by PCR amplification with target plasmids as templates, while the truncated and processed dsDNA targets were made through oligo annealing in 1× NEBuffer™ r2.1 buffer (50 mM NaCl, 10 mM Tris-HCl, 10 mM MgCl₂, 100 μg/ml Recombinant Albumin, pH 7.9. New England Biolabs, cat. #B6002). Forward (non-target strand, NTS) and reverse (target strand, TS) oligos were annealed at a 1:1 molar ratio by heating at 95 °C for 2 minutes and gradually cooling to 4 °C over 30 min using a thermocycler. Reverse oligos were used when evaluating the performance of ssDNA target. Purified protein BhCas12b/DpbCas12e,

in vitro transcribed sgRNA or tracrRNA-crRNA or Rptr-mRNA, target DNA fragment, and single-strand DNA reporter (CJpr1234, 5′-FAM-TTATT-3′-IABkFQ) were added into 1× NEBuffer™ r2.1 buffer at a final concentration of 100 nM, 100 nM, 10 nM and 1000 nM, respectively. The trans-cleavage assay was performed in 5-μl reactions by measuring fluorescence on a Synergy Neo2 plate reader (BioTek) using a 96-well V-bottom plate (Corning Costar, cat. #3357) with an excitation filter of 485 nm and an emission filter of 528 nm. Time-courses were run for 16 h with an interval of 3 min between reads. Observed rates ($k_{obs}$) were obtained by fitting the kinetic traces to a one-phase association exponential function using Prism 9 (GraphPad Software, Inc.).

To compare background activities between sgRNA and Rptr, purified protein BhCas12b, in vitro transcribed sgRNA or Rptr/sensed RNA, target DNA, and single-strand DNA reporter (CJpr1234) were added into 1× NEBuffer™ r2.1 buffer at a final concentration of 100 nM, 100 nM, 10 nM and 1,000 nM, respectively. The absence of the dsDNA target served as the negative control for the sgRNA setting. The absence of the sense RNAs served as the negative control for the Rptr setting. Values at 2 h and 16 h were used to make plots.

For the guide:target mismatch tolerance test, purified protein (BhCas12b, BthCas12b or AacCas12b), in vitro transcribed Rptrs, sense RNAs, dsDNA target variants, and single-strand DNA reporter (MGhD001, 5′-FAM-TTTTT-3′-IABkFQ) were added into 1× NEBuffer™ r2.1 buffer at a final concentration of 100 nM, 50 nM, 50 nM, 10 nM and 1,000 nM, respectively. For Rptr:sensed RNA mismatch tolerance test, purified protein BthCas12b, in vitro transcribed Rptr, sense RNA variants, dsDNA target, and single-strand DNA reporter (MGhD001, 5′-FAM-TTTTT-3′-IABkFQ) were added into 1× NEBuffer™ r2.1 buffer at a final concentration of 150 nM, 300 nM, 100 nM, 10 nM and 1,000 nM, respectively. Values at 1 h, 2 h and 16 h were used to make plots.

For 16 S rRNA detection of multiple pathogens, purified protein BthCas12b, one universal Rptr, 16 S rRNA fragment of different pathogens, target DNA fragments, and a single-strand DNA reporter (MGhD001) were added into 1× NEBuffer™ r2.1 buffer at a final concentration of 150 nM, 300 nM, 100 nM, 10 nM and 1000 nM, respectively.

## Statistics and reproducibility

All statistical comparisons were performed using a Student's T test assuming unequal variance. Comparisons were only performed with at least three biological replicates. A one-tailed test was used in all instances based on the null hypothesis that the sample exhibited equal or worse performance than the control. Values were assumed to be normally distributed. The threshold of significance for the calculated *p*-values was set as 0.05. Sample sizes were selected to align with prior work. No data were excluded from the analyses with the exception of individual TXTL runs yielding no fluorescence above background; these instances are marked with a dash in the Source Data. The experiments were not randomized. The Investigators were not blinded to allocation during experiments and outcome assessment.

## Reporting summary

Further information on research design is available in the Nature Portfolio Reporting Summary linked to this article.

# Data availability

All data related to work are provided with this paper as Source Data. Crystal structures used in this work include PDB 7VYX, 5WTI, 5WQE, 8J12 and 6NY3. Source data are provided with this paper.

# Code availability

No custom code was used in this work.

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

## Acknowledgements

pYPQ290 (Addgene plasmid # 129670) and pYPQ291 (Addgene plasmid # 129671) were gifts from Yiping Qi; pZ149-pcDNA3-BhCas12b_v4 was a gift from Feng Zhang (Addgene plasmid # 122446); pET28a-SUMO-AsCas12f was a gift from Quanjiang Ji (Addgene plasmid # 171612); pCasX1 was a gift from Jennifer Doudna (Addgene plasmid # 87685). Wulf Blankenfeldt kindly provided purified PbuCas13b. We thank the recombinant protein expression facility of the Rudolf-Virchow-Center in Würzburg for expression and purification of BhCas12b, BthCas12b, AacCas12b and DpbCas12e. This work was supported by ERC Consolidator grant (865973 to C.L.B.) and the GO-Bio Initial programme of the Bundesministerium für Bildung und Forschung (031B0989 and 16LW0132 to D.C. and C.L.B.).

## Author contributions

Conceptualization: C.J. and C.L.B.; Methodology: C.J., N.L.P, J.Y., M.G.M, D.C. and C.L.B.; Formal Analysis: C.J., N.L.P, J.Y., M.G.M, D.C. and C.L.B.; Investigation: C.J., N.L.P, J.Y., M.G.M, D.C., S.K., S.L.M. and R.L.; C.J., N.L.P and J.Y. performed the TXTL assay for BhCas12b and DpbCas12e; C.J., J.Y. and S.L.M. performed the TXTL assay for AsCas12f1. C.J. and J.Y. performed the collateral assay for BhCas12b and DpbCas12e, and sgRNA versus Rptr background activity analysis for BhCas12b; M.G.M. and R.L. performed the targeting specificity analysis for three Cas12b orthologs and 16 S rRNA detection. S.K. performed mismatches tolerance test in the repeat:anti-repeat duplex for BhCas12b and compared PUMA with DETECTR and SHERLOCK. D.C. performed the TXTL assay for BthCas12b. Writing – Original Draft: C.J. and C.L.B.; Writing – Review and Editing: C.J., N.L.P, J.Y., M.G.M, D.C., S.L.M., R.L. and C.L.B.; Visualization: C.J., and C.L.B.; Supervision: C.L.B.; Funding Acquisition: D.C. and C.L.B.

## Funding

## Competing interests

Provisional patent applications have been filed to the European Patent Office on concepts related to Cas12 tracrRNA reprogramming and DNA target truncations by C.J. and C.L.B. (application numbers WO2021170877A1, EP23200244.4). C.L.B. is a co-founder of Leopard Biosciences, a co-founder and Scientific Advisory Board member of Locus Biosciences, and is a Scientific Advisory Board member of Benson Hill. The other authors declare no competing interests.
