## [Peer Review File · Nature Communications]

REVIEWER COMMENTS

Reviewer #1 (Remarks to the Author):

This work authored by Jiao et al. introduced an approach for PAM-free detection of RNA through flexibly reprogramming the anti-repeat sequence of tracrRNA to make it complementary (or with several mismatches) to the repeat of crRNA containing the RNA sequence to be detected, therefore activates the functionality of crRNA and triggers the collateral activity, resulting in indiscriminate degradation of ssDNA-FQ for an indication of successful detection. This approach is potentially interesting, as it converts the detection target from a sequence complementary to the guide to a sequence in repeat of crRNA complementary to the anti-repeat of tracrRNA. Also, there is a profound meaning in investigating how the anti-repeat can be designed to induce the functionality of the guide. This paper is overall clearly written, despite some points that need to be further clarified. Below are some major concerns primarily regarding the experimental design and article structure organization.

1. The specificity in recognizing nucleotide mismatches is important for evaluation of a nucleic acid detection assay. I encourage the authors to investigate the situations where there are several base mismatches between the repeat of crRNA and the anti-repeat of tracrRNA. I noticed that in Fig. 6b the designed anti-repeat has one mismatch to the long repeat of *S. aureus*, *E. faecalis* and *L. monocytogenes* at the fourth base counted from the 3' end of tracrRNA and this assay seemed to still work properly. I am wondering if more mismatches are tolerated. I suggest the authors to start from one mismatch and successively add one mismatch until a significant suppression of trans-nuclease activity is observed. I also suggest the authors to design mismatches in both the LR/AR and SR/AR region, as well as the triple helix and its surrounding region for DpbCas12e.
2. In the section describing the reduction of background fluorescence using Rptrs, the authors need to clarify the blank control for testing with sgRNAs and Rptrs. In my understanding, the blank control with sgRNAs was a group without dsDNA target, and the blank control with Rptrs was a group without Rptrs and dsDNA target. In order to prove the author's hypothesis that "we hypothesize any background activities would be greatly reduced using Rptrs, as the guide RNA is principally formed only in the presence of the sensed RNA", I suggest the authors perform a control experiment in the presence of Rptrs but without the dsDNA target. If the hypothesis is true, the background fluorescence will be recovered under this situation. Also, I suggest the authors to use multiple guide sequences (perhaps more than 4) to verify if this high background is a stable phenomenon with different guide sequences.
3. The authors intensively study the truncation of dsDNA target to improve the trans-cleavage rate, which is fantastic. However, I believe it is more meaningful to study how the sequence of anti-repeats in tracrRNA will affect the trans-cleavage rate, as the sequence encoded in Rptrs is the real target they want to detect. Meanwhile, I think the hybridization kinetics between crRNA and tracrRNA worths a thorough investigation as in some cases, the homodimers in crRNA and tracrRNA may greatly circumvent the formation of crRNA/tracrRNA heterodimer and I believe this is the real rate-limiting step of this assay. I suggest the authors design crRNAs or tracrRNAs with different secondary structures and

investigate the effect on trans-cleavage rates. Moreover, I suggest the authors compare the trans-cleavage rates between using sgRNA and Rptrs to see if an additional hybridization step will reduce the efficiency.

4. As the trans-cleavage was still activated through the guide recognizing a corresponding dsDNA target, the sensitivity of PUMA should be approximate to that of a conventional trans-cleavage assay with a regular crRNA or sgRNA. As the authors showed in Fig. 5c, the sensitivity should be at around 100 pM at best. I noticed in their pathogen detection section they used 100 nM of 16S rRNA as a synthetic target for detection, this concentration is apparently too high in practical scenarios. I am wondering if this sensitivity is enough to detect real-world samples or clinical samples. In addition, the assay time seems to take too long (2~16h), which is much longer than conventional CRISPR-based assay (e.g., less than 1 h).

5. In Fig. 1, I understand the authors intention to compare PUMA to the conventional DETECTR assay and the necessity to introduce the classification of hybridization situation of crRNA and tracrRNA for different Cas enzyme orthologs. However, I am not clear about the need to present Fig. 1b here. In my opinion, it is of little relationship to the general aim of the first section of this study in terms of introducing tracrRNA reprogramming. Can the authors explain the need to display this figure here? Moreover, since the authors are trying to explore the opportunity of reprogramming tracrRNA for Cas12 orthologs, I think it is not necessary to introduce the duplex structure of crRNA/tracrRNA for Cas9 orthologs in Fig. 1c, or the authors should consider move them to the SI.

6. It appears that the long sequence of the Rptrs region containing LR:AR needs to be complementary to the target RNA. In these cases, the target RNA that can be detected is thought to be very limited. If the Rptr region is not bound to the target RNA (sensed RNA), can it still function the same?

7. Have the authors tested with ssDNA instead of dsDNA target in Figure 4a?

Some minor points:

1. Line 225, "stand" should be "strand".
2. Line 227, "Complete cleavage of the dsDNA target is a prerequisite of collateral cleavage". How do you explain the background fluorescence when there is no dsDNA target present?
3. Line 232-238, Fig. S4d does not match the content. Should that be Fig. S5a or Fig. S5b instead?
4. Line 723, "four different duplexes", but there are five displayed in Fig. 1c.

Reviewer #2 (Remarks to the Author):

In this manuscript, Jiao, et al. present a novel system to sense RNAs using Cas12 nucleases. While this may seem counterintuitive (Cas12 recognizes dsDNA), the authors make use over the complexity of the Cas12 tracrRNA to link the presence of a desired RNA target to collateral dsDNA cleavage. To do this, they convert the RNA target into a functional gRNA via the introduction of a designed Rptr RNA, which then creates a functional scaffold for Cas12 loading and subsequent target DNA recognition. The authors demonstrate the simplicity and programmability of this approach on multiple different Cas12s, including those with very complex WT tracrRNA structures, and those with triple helical RNA scaffolds. Using their previously-designed TXTL-based assays, a good proxy for diagnostic-like platforms, they demonstrate effective dsDNA cleavage and make the obvious jump to activate collateral Cas12-based ssDNA cleavage for signal visualization and amplification, as was done with DETECTR. They then design a universal Rptr RNA that can detect different 16S ribosomal RNAs, as these RNAs have a universal conserved region. Most importantly, the sensing is done in a PAM-independent manner, though it is presumed that the dsDNA target requires a PAM for cleavage. Nonetheless, the fact that the actual sensed RNA does not require a PAM makes the system very modular.

Overall, this is a very strong, thorough paper that comprehensively engineers Rptr RNAs for RNA detection and dsDNA cleavage by diverse Cas12s, extending the applicability of Cas12-based diagnostics systems. However, to make the paper stronger, I recommend a few additional experiments and clarifications:

Major Concerns

1. RNA sensing for signal amplification in a PAM-independent manner has been achieved via Cas13-based systems. For at least 3 target RNAs, the authors should do a side-by-side comparison of all designed PUMA systems with a single SHERLOCK-based Cas13 system. This should be done with all of the possible Cas12 orthologs that have designed Rptrs. Even if the authors do not observe superior RNA sensing and signal amplification, it would be a good benchmark. One of the targets should have a requisite PAM for a DETECTR system, so that a direct DETECTR vs. SHERLOCK vs. PUMA comparison can be made.
2. Perhaps I missed this, but it would be good to clarify the design principles of the target dsDNA and sensed RNA sequence for maximum signal amplification. Doing a thorough testing (10-12) of additional RNA targets with distinct sequence composition would help readers appreciate the modularity of the system.

Minor Concerns

1. I would appreciate if the authors were more clear on the PAM-independent claim. From a first read of the text, it seems like the Cas12 enzyme no longer needs a PAM for dsDNA cleavage, when in fact it is the RNA sensing that is PAM-independent. The target dsDNA still needs a PAM, I presume. I would clarify the language in the abstract and introduction to make this clear.

Reviewer #3 (Remarks to the Author):

The authors develop a Cas12-based nucleic acid diagnostic that cleaves a fluorophore-quencher pair to generate fluorescence in response to the presence of a specific RNA transcript sequence. In contrast to prior efforts, the diagnostic uses a modified tracrRNA (called a Rptr) to specifically bind to a targeted RNA sequence in complex with a guide RNA that binds to a targeted DNA strand. When Cas12 cleaves a targeted DNA strand, it also activates promiscuous cleavage activity of single-stranded DNA, which is leveraged by the authors' diagnostic to cleave a fluorophore-quencher pair and generate fluorescence. The key difference between the authors' diagnostic assay is prior use of Cas12 in other diagnostic assays is the use of the modified tracrRNA (Rptr) to provide more sequence specificity to target RNA sequences without a PAM motif requirement.

As part of the overall effort to develop this diagnostic assay, the authors carry out a comprehensive study to determine how several factors control the assay's background signal and RNA dosage response curve (sensitivity), using a cell-free expression platform as their assay environment. These factors include: [1] how differences in the Cas12 enzyme (BthCas12b, PmuCas12c1, DpbCas12e) alter the prerequisite tracrRNA:target RNA structure needed for proper loading up into the Cas12 enzyme; [2] how changes in the modified tracrRNA (Rptr) sequence alter its folding with the target RNA creates secondary & tertiary RNA structures, which alters its loading up into each respective Cas12, including nucleotide changes in a triple helix tertiary structure; and [3] how changes to the DNA target strand alters Cas12 cleavage & ssDNA cleavage rates, including shortening the DNA target strand and introducing staggered overhangs to accelerate promiscuous ssDNA cleavage activity. The authors demonstrate that the diagnostic assay can generate an RNA-specific fluorescence response with RNA sensitivities at around 100 pM to 1 nM and that the specificity is sufficient to adequately distinguish between five different 16S rRNAs from five different bacterial species.

Overall, while the study is carried out extremely well, the overall question tackled in the study (the development of another nucleic acid diagnostic) is very narrow and applied. The authors show that the diagnostic assay is not as sensitive to detecting sequence-specific RNA strands as compared to several prior efforts that also use Cas12 variants. In contrast to prior efforts in the field (including some by the corresponding author), this diagnostic assay can not be multiplexed in a one-pot reaction as it relies on promiscuous cleavage of ssDNA to generate a multiple turnover fluorescence response (which is discussed by the authors). While this new diagnostic assay does not require a PAM in the target RNA for

detection, the authors do not list applications where PAM-less detection of RNA is needed. The selected application (detecting bacterial pathogens) could have been accomplished via standard techniques (e.g. RT-qPCR or NGS) with greater precision and/or breadth. The authors do not discuss where this technology fits within the broader context of diagnostic assays and only compare it to prior efforts using Cas9 or Cas12 for RNA detection. For example, there is no discussion of assay costs and output modalities. The assay requires the addition of purified protein (a Cas12 variant) and the output is limited to fluorescence, which increases the overall cost of the diagnostic device, its reagents, and the necessary cold chain of the reagents. Therefore, a typical reader (not focused on CRISPR) will be left with several unanswered questions about the benefits of the developed diagnostic assay and how it improves upon the state-of-the-art. However, I think that a reader who studies CRISPR systems will be greatly interested in how changing the tracrRNA sequence alters the overall RNA complex's structure and its loading up into each Cas12 variant. They will also be interested in how changing the target DNA strand alters the transition towards promiscuous ssDNA cleavage activity.

RESPONSE TO REVIEWERS' COMMENTS

We thank the three reviewers for taking the time to carefully review our manuscript and for their constructive feedback that helped improve the overall manuscript. As part of addressing these comments, we made the following substantive experimental additions:

- Assessed the impact of mismatches in the repeat:anti-repeat duplex for BhCas12b (Fig. S14).
- Introduced additional controls when evaluating DNA target-independent collateral cleavage activity (Fig. S12).
- Screened additional sgRNAs for DNA target-independent collateral cleavage activity (Fig. S11).
- Assessed the impact of secondary structure within the Rptr binding region of the sensed RNA (Fig. S7).
- Assessed the impact of annealing on relative performance of sgRNA and sensed RNA-Rptr pair (Fig. S8).
- Assessed single-stranded DNA targets (Fig. S5).
- Compared PUMA to DETECTR and SHERLOCK (Fig. S13).

In addition to these changes, we calculated collateral cleavage rates (kobs) when assessing the impact of guide-target mismatches across the three Cas12b orthologs to make a fairer comparison. Figure 6a and the associated text in the Results section on p. 15 were updated accordingly.

Finally, we added a new author (Sarah Kono) who contributed to experiments related to Figs. S13 and S14.

Our response to each comment below is in blue text, while the associated major changes to the main text and SI are in red text.

Reviewer #1 (Remarks to the Author):

This work authored by Jiao et al. introduced an approach for PAM-free detection of RNA through flexibly reprogramming the anti-repeat sequence of tracrRNA to make it complementary (or with several mismatches) to the repeat of crRNA containing the RNA sequence to be detected, therefore activates the functionality of crRNA and triggers the collateral activity, resulting in indiscriminate degradation of ssDNA-FQ for an indication of successful detection. This approach is potentially interesting, as it converts the detection target from a sequence complementary to the guide to a sequence in repeat of crRNA complementary to the anti-repeat of tracrRNA. Also, there is a profound meaning in investigating how the anti-repeat can be designed to induce the functionality of the guide. This paper is overall clearly written,

despite some points that need to be further clarified. Below are some major concerns primarily regarding the experimental design and article structure organization.

We thank the reviewer for their helpful guidance. We address each raised major concern below, where the added data greatly improved the manuscript.

1. The specificity in recognizing nucleotide mismatches is important for evaluation of a nucleic acid detection assay. I encourage the authors to investigate the situations where there are several base mismatches between the repeat of crRNA and the anti-repeat of tracrRNA. I noticed that in Fig. 6b the designed anti-repeat has one mismatch to the long repeat of *S. aureus*, *E. faecalis* and *L. monocytogenes* at the fourth base counted from the 3' end of tracrRNA and this assay seemed to still work properly. I am wondering if more mismatches are tolerated. I suggest the authors to start from one mismatch and successively add one mismatch until a significant suppression of trans-nuclease activity is observed. I also suggest the authors to design mismatches in both the LR/AR and SR/AR region, as well as the triple helix and its surrounding region for DpbCas12e.

Following the reviewer's suggestion, we performed extensive mismatch testing within the LR/AR and SR/AR regions using BthCas12b given its use for 16S rRNA detection. As shown below, we tested between one and four consecutive mismatches in either region. All mismatches except those at M21-23 led to signal production with intensities ranging from 44% to 94% of their unmodified counterpart. The tolerance of mismatches in the Rptr binding region lends to the development of the universal primer used to probe 16S rRNA sequences, although it can lead to the generation of unintended ncrRNAs.

These new data were incorporated as Figure S14 and Table S1 Tab2 and described with the following in the Results section on p. 15:

“We also evaluated the extent to which BthCas12b accepts mismatches between the sensed RNA and the Rptr (Fig. S14). Mismatches in the long or short repeat consistently and significantly reduced but rarely eliminated activation of collateral cleavage activity even with four consecutive mismatches.”

The reviewer also suggested testing mutations in the triple helix for DpbCas12e. Given our extensive mutational analysis (Fig. 3g) representing over five times as many variants as that tested in the original characterization of the DpbCas12e tracrRNA (doi: 10.1038/s41586-019-0908-x), we felt this analysis would be a sufficient indicator of non-canonical sequences.

2. In the section describing the reduction of background fluorescence using Rptrs, the authors need to clarify the blank control for testing with sgRNAs and Rptrs. In my understanding, the blank control with

sgRNAs was a group without dsDNA target, and the blank control with Rptrs was a group without Rptrs and dsDNA target. In order to prove the author’s hypothesis that “we hypothesize any background activities would be greatly reduced using Rptrs, as the guide RNA is principally formed only in the presence of the sensed RNA”, I suggest the authors perform a control experiment in the presence of Rptrs but without the dsDNA target. If the hypothesis is true, the background fluorescence will be recovered under this situation. Also, I suggest the authors to use multiple guide sequences (perhaps more than 4) to verify if this high background is a stable phenomenon with different guide sequences.

Following the reviewer’s suggestion, we performed two experiments related to DNA target-independent collateral activity: (1) introduce more controls around Rptrs and (2) test a larger panel of sgRNAs. In the first experiment, we assessed all combinations of the PUMA components for sensed RNA1 whose sgRNA yielded high DNA target-independent collateral activity. As shown below, the Rptr plus the sensed RNA—but not either component alone—drove collateral activity even in the absence of the DNA target, in line with the results with an sgRNA and directly supporting the hypothesis.

These results were integrated as Figure S12 and noted with the following on p. 14:

“Supporting this hypothesis, combining a sensed RNA and Rptr for BhCas12b drove collateral activity even in the absence of the DNA target (Fig. S12).”

For the second experiment, we tested 10 additional sgRNAs encoding unique guide sequences, with five targeting different locations in the *CJ8421_04975* mRNA (#5-9) and the rest targeting different locations in the deGFP mRNA (#1-5). We included two prior sgRNAs targeting the *CJ8421_04975* mRNA (#1, 4) as examples of high and low background collateral activity. As shown below, two of the new sgRNAs exhibited leaky collateral activity in the absence of a DNA

target substantially exceeding that of the low-activity control, reinforcing the prevalence of this phenomenon.

The new data were incorporated as Figure S11, and we added the following to p. 13:

“This phenomenon was not isolated, as 2 out of 10 additional sgRNAs we tested exhibited DNA target-independent collateral activity that was significantly higher than that of sgRNA#4 exhibiting low background activity (Fig. S11).”

3. The authors intensively study the truncation of dsDNA target to improve the trans-cleavage rate, which is fantastic. However, I believe it is more meaningful to study how the sequence of anti-repeats in tracrRNA will affect the trans-cleavage rate, as the sequence encoded in Rptrs is the real target they want to detect. Meanwhile, I think the hybridization kinetics between crRNA and tracrRNA worths a thorough investigation as in some cases, the homodimers in crRNA and tracrRNA may greatly circumvent the formation of crRNA/tracrRNA heterodimer and I believe this is the real rate-limiting step of this assay. I suggest the authors design crRNAs or tracrRNAs with different secondary structures and investigate the effect on trans-cleavage rates. Moreover, I suggest the authors compare the trans-cleavage rates between using sgRNA and Rptrs to see if an additional hybridization step will reduce the efficiency.

We agree that multiple factors could influence the extent of collateral cleavage activity, with the secondary structure of the sensed RNA as one factor. To probe this factor directly, we introduced complementary sequences of different lengths to the 5' end of an RNA sensed by the BthCas12b, resulting in different predicted hairpin lengths. As shown below, introducing a hairpin either adjacent to or incorporating the anti-repeat region reduced the rate of collateral cleavage activity, with increased inhibition with larger hairpins. These results show that secondary is a separate consideration that should be taken into account and studied in greater detail when selecting potential binding sites for Rptrs.

The new data were incorporated as Figure S7, and we added the following to p. 12:

“To test this factor directly, we introduced extensions to the 5' extensions to the ncrRNA associated with BhRptr4 (Fig. S7). The hairpins reduced collateral cleavage activity, with an internal hairpin inhibiting more strongly than a flanking hairpin.”

The reviewer also noted that differences in collateral-cleavage activity between an ncrRNA-Rptr pair and an sgRNA could indicate other bottlenecks in RNA sensing. Using hybridization of the ncrRNA-Rptr pair as one important example, we compared the targeting activity of a representative sensed RNA-Rptr pair (a portion of *CJ8421_04975* mRNA and Rptr4) of BthCas12b with or without pre-annealing. Surprisingly, this pair outperformed the equivalent sgRNA, providing one example where sensed RNA-Rptr hybridization does not impose an obvious impediment.

The new data were incorporated as Figure S8, and we added the following to p. 12:

“In the absence of these structures though, introducing an annealing step did not enhance collateral cleavage activity (Fig. S8). Surprisingly, collateral cleavage activity resulting from pairing of partial CJ8421_04975 mRNA fragment and BhRptr4 was higher than that obtained with the equivalent sgRNA, indicating that hybridization between a sensed RNA and Rptr is not necessarily a bottleneck.”

4. As the trans-cleavage was still activated through the guide recognizing a corresponding dsDNA target, the sensitivity of PUMA should be approximate to that of a conventional trans-cleavage assay with a regular crRNA or sgRNA. As the authors showed in Fig. 5c, the sensitivity should be at around 100 pM at best. I noticed in their pathogen detection section they used 100 nM of 16S rRNA as a synthetic target for detection, this concentration is apparently too high in practical scenarios. I am wondering if this sensitivity is enough to detect real-world samples or clinical samples. In addition, the assay time seems to take too long (2~16h), which is much longer than conventional CRISPR-based assay (e.g., less than 1 h).

We agree 100 nM would greatly exceed the expected concentration of a sensed RNA in most applications. However, Cas12- and Cas13-based diagnostics are normally combined with an isothermal pre-amplification step, resulting in vastly increased concentrations of the sensed nucleic acid. For instance, in the original report of SHERLOCK (doi: 10.1126/science.aam9321), pre-amplification increased the sensitivity of RNA detection by 10^8 -fold from ~500 pM down to ~5 aM. We now note this higher concentration mimicking a pre-amplification step with the following on p. 16:

“The fragment was introduced at a final concentration of 100 nM, reflecting the output of isothermal pre-amplification and *in vitro* transcription².”

The reviewer also notes the longer timescales of our reactions, which would exceed a reasonable timeframe for a diagnostic test. For this work, we selected these timescales to capture the full range of collateral cleavage activities, such as when evaluating the different DNA target lengths in Figure 4 or assessing target-independent collateral cleavage activity in Figure 5. The lower timeframe of 2 hours also falls in the range reported in the original report of SHERLOCK (doi: 10.1126/science.aam9321). That stated, we assessed earlier time points as part of 16S rRNA detection in Figure 6. As shown below, values after 36-minute reactions were sufficient to statistically differentiate the different pathogens.

These data were added as Figure 6c, and the displayed 2-hour data were incorporated as Fig. S15.

5. In Fig. 1, I understand the authors intention to compare PUMA to the conventional DETECTR assay and the necessity to introduce the classification of hybridization situation of crRNA and tracrRNA for different Cas enzyme orthologs. However, I am not clear about the need to present Fig. 1b here. In my opinion, it is of little relationship to the general aim of the first section of this study in terms of introducing tracrRNA reprogramming. Can the authors explain the need to display this figure here? Moreover, since the authors are trying to explore the opportunity of reprogramming tracrRNA for Cas12 orthologs, I think it is not necessary to introduce the duplex structure of crRNA/tracrRNA for Cas9 orthologs in Fig. 1c, or the authors should consider move them to the SI.

Our intent with Figure 1b was to illustrate that many (but not all) type V CRISPR-Cas subtypes encode a tracrRNA. As most people working on CRISPR technologies are familiar with Cas12a nucleases that do not require a tracrRNA, we felt this illustration was important to inform this broad group of potential readers. From this large set, we chose three subtypes (V-B, V-E, V-F) to probe more directly in this work.

Figure 1c was also important to illustrate the diversity of repeat:anti-repeat duplexes associated with the different subtypes. Not only are these structures more complicated than those associated with type II repeat:anti-repeat duplexes, but they also vary in their composition from a single pseudoknot to multiple pseudoknots and even a triple helix structure. This complexity and

variability create a greater challenge for tracrRNA reprogramming than we faced with Type II systems, which we wanted to emphasize in the opening figure of the manuscript.

6. It appears that the long sequence of the Rptrs region containing LR:AR needs to be complementary to the target RNA. In these cases, the target RNA that can be detected is thought to be very limited. If the Rptr region is not bound to the target RNA (sensed RNA), can it still function the same?

Regarding the need for the LR:AR region to be complementary to the target RNA, we view this as an additional opportunity for target specificity rather than a limitation of the technology. The Rptr is designed to base pair with this additional region of the sensed RNA, allowing the Cas12 nuclease to recognize it as a guide RNA. Because the design rules afford ample flexibility in the duplex sequence, almost any sequence can be used—albeit with the exception of the triple helix structure in the RNA duplex of Cas12e. We further show that this RNA duplex is required, as Rptrs that do not form the full duplex with the sensed RNA fail to elicit collateral cleavage activity (see Figs. 2d-e and 3b-h). The only notable limitation is that sensed RNAs must be sufficiently long to accommodate the guide and Rptr binding region, which we now note with the following on p. 16-17:

“Virtually any RNA sequence can be detected with PUMA generally as long as the RNA sequence follows the elucidated design rules (Fig. S3) and meets a minimum length comprising an anti-repeat and guide.”

7. Have the authors tested with ssDNA instead of dsDNA target in Figure 4a?

We originally did not test single-stranded DNA as a target due to the heightened mismatch tolerance reported previously (doi: 10.1126/science.aar6245). However, as this substrate could accelerate activation of collateral cleavage activity due to circumventing PAM recognition and DNA helicase activity, we compared the use of ssDNA and dsDNA targets. As shown below, we found that the ssDNA targets consistently exhibited higher collateral cleavage activity compared to dsDNA targets of the equivalent size.

We incorporated the new data as Figure S5, and we added the following to p. 11:

“We also tested ssDNA targets, which exhibited at least a 2-fold increase in collateral activity than dsDNA targets of equivalent size (Fig. S5), in line with circumventing PAM recognition and DNA unwinding.”

Some minor points:

1. Line 225, “stand” should be “strand”.

We have corrected this typographical error.

2. Line 227, “Complete cleavage of the dsDNA target is a prerequisite of collateral cleavage”. How do you explain the background fluorescence when there is no dsDNA target present?

The reviewer rightfully notes that our observation of collateral cleavage activity in the absence of a provided DNA target conflicts with this statement. As the underlying mechanism of this collateral activity remains elusive and worth a separate study, we have rephrased this sentence on p. 11 to read the following:

“Complete cleavage of the dsDNA target normally precedes collateral cleavage....”

3. Line 232-238, Fig. S4d does not match the content. Should that be Fig. S5a or Fig. S5b instead?

We thank the reviewer for pointing out this error. This should be Fig. S6a-b instead of Fig. S4d in the revised manuscript. We have updated the main text accordingly.

4. Line 723, “four different duplexes”, but there are five displayed in Fig. 1c.

Another excellent catch by the reviewer. We have changed this statement to five different duplexes.

Reviewer #2 (Remarks to the Author):

In this manuscript, Jiao, et al. present a novel system to sense RNAs using Cas12 nucleases. While this may seem counterintuitive (Cas12 recognizes dsDNA), the authors make use over the complexity of the Cas12 tracrRNA to link the presence of a desired RNA target to collateral dsDNA cleavage. To do this, they convert the RNA target into a functional gRNA via the introduction of a designed Rptr RNA, which then creates a functional scaffold for Cas12 loading and subsequent target DNA recognition. The authors demonstrate the simplicity and programmability of this approach on multiple different Cas12s, including those with very complex WT tracrRNA structures, and those with triple helical RNA scaffolds. Using their previously-designed TXTL-based assays, a good proxy for diagnostic-like platforms, they demonstrate effective dsDNA cleavage and make the obvious jump to activate collateral Cas12-based ssDNA cleavage for signal visualization and amplification, as was done with DETECTR. They then design a universal Rptr RNA that can detect different 16S ribosomal RNAs, as these RNAs have a universal conserved region. Most importantly, the sensing is done in a PAM-independent manner, though it is presumed that the dsDNA target requires a PAM for cleavage. Nonetheless, the fact that the actual sensed RNA does not require a PAM makes the system very modular.

Overall, this is a very strong, thorough paper that comprehensively engineers Rptr RNAs for RNA detection and dsDNA cleavage by diverse Cas12s, extending the applicability of Cas12-based diagnostics systems. However, to make the paper stronger, I recommend a few additional experiments and clarifications:

We thank the reviewer for this supportive overview and for their encouragement. We address each point below and in doing so have strengthened the overall work.

Major Concerns

1. RNA sensing for signal amplification in a PAM-independent manner has been achieved via Cas13-based systems. For at least 3 target RNAs, the authors should do a side-by-side comparison of all designed PUMA systems with a single SHERLOCK-based Cas13 system. This should be done with all of the possible Cas12 orthologs that have designed Rptrs. Even if the authors do not observe superior RNA sensing and signal amplification, it would be a good benchmark. One of the targets should have a requisite PAM for a DETECTR system, so that a direct DETECTR vs. SHERLOCK vs. PUMA comparison can be made.

The reviewer raises an excellent point about benchmarking PUMA with the existing DETECTR and SHERLOCK platforms. We therefore directly compared the detection sensitivity of PUMA, DETECTR, and SHERLOCK on equal footing, but we left out the pre-amplification step to better gauge sensitivity. BhCas12b was used for DETECTR to provide a direct comparison to PUMA, while PbuCas13b was used with SHERLOCK. In each case, we varied the concentration of the sensed substrate (RNA for PUMA and SHERLOCK, dsDNA for DETECTR) and kept all other components constant. Three new loci (Locus 1-3) from the *CJ8421_04975* gene were selected to provide overlapping targeting sites for the three platforms.

For DETECTR, sgRNA1 and sgRNA3 showed background activity and lacked the consensus PAM sequence upstream of their matched DNA targets. Therefore, locus 2 was chosen for benchmarking all three platforms. The limit of detection for DETECTR with BhCas12b was approximately 10 nM, worse than that of PUMA with BhCas12b and SHERLOCK with PbuCas13b (1 nM). At loci 1 and 3, either PUMA or SHERLOCK outperformed the other. These comparisons indicate that PUMA can perform similarly to DETECTR and SHERLOCK in detection sensitivity, at least for the selected Cas nucleases. Given that PUMA can directly detect RNA, is not restricted to specific PAMs flanking the sensed sequence, and can use more affordable and stable ssDNA molecular beacons, it can hold notable advantages over DETECTR and SHERLOCK.

The new data were incorporated as Figure S13, with the accompanying description in the Results section on p. 14.

2. Perhaps I missed this, but it would be good to clarify the design principles of the target dsDNA and sensed RNA sequence for maximum signal amplification. Doing a thorough testing (10-12) of additional RNA targets with distinct sequence composition would help readers appreciate the modularity of the system.

Our performed work allowed us to elucidate general rules for Rptr design, although extending these rules to maximize signal amplification has typically required testing much larger gRNA-target sets followed by machine learning (e.g., doi: 10.1038/s41587-022-01213-5). As such efforts

require many more sensed RNA-Rptr pairs and represent entire studies in themselves, sufficiently expanding the set of gRNA-targets is better left to follow-on work. We also note that, in the course of this work and the revisions, we tested 15 RNA targets (including 3 additions from the revision experiments) across TXTL and *in vitro* assays using 5 different Cas12 nucleases, 17 Rptrs and 66 modified sgRNAs. Based on these insights, we previously provided an illustration of the current design rules in Fig. S3. As part of the revision, we updated the Methods section on p. 27 to provide a clearer understanding of the design approach.

Minor Concerns

1. I would appreciate if the authors were more clear on the PAM-independent claim. From a first read of the text, it seems like the Cas12 enzyme no longer needs a PAM for dsDNA cleavage, when in fact it is the RNA sensing that is PAM-independent. The target dsDNA still needs a PAM, I presume. I would clarify the language in the abstract and introduction to make this clear.

We had used the phrase “PAM-independent” to parallel other studies using the same phrasing (e.g., doi: 10.1093/nar/gkac1144) or “PAM-free” (e.g., doi: 10.1016/j.ab.2023.115046). While we agree such phrasing can incorrectly insinuate that no PAM is required at any step, we aimed to only use the phrase in the context of RNA detection—the step in which no PAM is required. We therefore reviewed every instance in which we used “PAM-independent” and rephrased where necessary to avoid making this insinuation. For instance, we rephrased the first sentence in the abstract noting the lack of a PAM to the following:

“Here, we reprogram tracrRNAs from diverse Cas12 nucleases, linking the presence of an RNA-of-interest to dsDNA cleavage and subsequent collateral single-stranded DNA cleavage—all without the RNA necessarily encoding a protospacer-adjacent motif (PAM).”

We also rephrased the first description of PAM-independent detection on p. 3 of the Introduction with the following to more clearly state our intent with this phrase:

“Tackling the first restriction, multiple studies have recently reported PAM-independent detection of dsDNA in which the detected target sequence was not flanked by a PAM.”

Reviewer #3 (Remarks to the Author):

The authors develop a Cas12-based nucleic acid diagnostic that cleaves a fluorophore-quencher pair to generate fluorescence in response to the presence of a specific RNA transcript sequence. In contrast to prior efforts, the diagnostic uses a modified tracrRNA (called a Rptr) to specifically bind to a targeted RNA sequence in complex with a guide RNA that binds to a targeted DNA strand. When Cas12 cleaves a targeted DNA strand, it also activates promiscuous cleavage activity of single-stranded DNA, which is leveraged by the authors' diagnostic to cleave a fluorophore-quencher pair and generate fluorescence. The key difference between the authors' diagnostic assay is prior use of Cas12 in other diagnostic assays is the use of the modified tracrRNA (Rptr) to provide more sequence specificity to target RNA sequences without a PAM motif requirement.

As part of the overall effort to develop this diagnostic assay, the authors carry out a comprehensive study to determine how several factors control the assay's background signal and RNA dosage response curve (sensitivity), using a cell-free expression platform as their assay environment. These factors include: [1] how differences in the Cas12 enzyme (BthCas12b, PmuCas12c1, DpbCas12e) alter the prerequisite tracrRNA:target RNA structure needed for proper loading up into the Cas12 enzyme; [2] how changes in the modified tracrRNA (Rptr) sequence alter its folding with the target RNA creates secondary & tertiary RNA structures, which alters its loading up into each respective Cas12, including nucleotide changes in a triple helix tertiary structure; and [3] how changes to the DNA target strand alters Cas12 cleavage & ssDNA cleavage rates, including shortening the DNA target strand and introducing staggered overhangs to accelerate promiscuous ssDNA cleavage activity. The authors demonstrate that the diagnostic assay can generate an RNA-specific fluorescence response with RNA sensitivities at around 100 pM to 1 nM and that the specificity is sufficient to adequately distinguish between five different 16S rRNAs from five different bacterial species.

Overall, while the study is carried out extremely well, the overall question tackled in the study (the development of another nucleic acid diagnostic) is very narrow and applied. The authors show that the diagnostic assay is not as sensitive to detecting sequence-specific RNA strands as compared to several prior efforts that also use Cas12 variants. In contrast to prior efforts in the field (including some by the corresponding author), this diagnostic assay can not be multiplexed in a one-pot reaction as it relies on promiscuous cleavage of ssDNA to generate a multiple turnover fluorescence response (which is discussed by the authors). While this new diagnostic assay does not require a PAM in the target RNA for detection, the authors do not list applications where PAM-less detection of RNA is needed. The selected application (detecting bacterial pathogens) could have been accomplished via standard techniques (e.g. RT-qPCR or NGS) with greater precision and/or breadth. The authors do not discuss where this technology fits within the broader context of diagnostic assays and only compare it to prior efforts using Cas9 or Cas12 for RNA detection. For example, there is no discussion of assay costs and output modalities. The assay requires the addition of purified protein (a Cas12 variant) and the output is limited to fluorescence, which increases the overall cost of the diagnostic device, its reagents, and the necessary cold chain of the reagents. Therefore, a typical reader (not focused on CRISPR) will be left with several unanswered questions about the benefits of the developed diagnostic assay and how it improves upon the state-of-the-art. However, I

think that a reader who studies CRISPR systems will be greatly interested in how changing the tracrRNA sequence alters the overall RNA complex's structure and its loading up into each Cas12 variant. They will also be interested in how changing the target DNA strand alters the transition towards promiscuous ssDNA cleavage activity.

We thank the reviewer for their on-point overview of our work and balanced efforts to tie it into broader work within the field—including our own published studies. The reviewer raised a few critiques, which we address in turn below. Overall, we believe the first demonstration of Cas12 tracrRNA reprogramming will already appeal to the broad audience in CRISPR biology and technologies, while demonstrating PAM-independent RNA detection provides a practical output. At the same time, addressing the points below helped broaden our work.

General goal of the study. The primary directive of the work was showing for the first time how Cas12 tracrRNAs could be reprogrammed to convert sensed RNAs into gRNAs. The conversion was more challenging than we had previously encountered with Cas9 tracrRNAs due to the presence of pseudoknots and triple helix structures. However, through systematic analyses, we found that these tracrRNAs could be readily reprogrammed following simple design rules. Unlike Cas9, Cas12 introduced collateral cleavage that could boost the signal. This became the impetus to investigate RNA detection. Through this work, we further discovered that truncating the DNA target can accelerate collateral cleavage, while PUMA avoids DNA target-independent collateral cleavage associated with some BhCas12b gRNAs.

Sensitivity compared to other CRISPR-based diagnostic approaches. Paralleling our response to Major Comment #1 from Reviewer #2, we directly compared PUMA to the two standard CRISPR-based diagnostic platforms DETECTR (for DNA) and SHERLOCK (for RNA). We direct the reviewer to this comment for more information.

Potential of multiplexing similar to LEOPARD. We agree that relying on non-specific collateral cleavage inherently undercuts the multiplexability that came with LEOPARD. However, through a paragraph in the discussion, we proposed different schemes where multiplexability could be regained, such as printing DNA targets at individual spots on microarrays or encapsulating DNA targets in microfluidic droplets. We have slightly revised the relevant sentence proposing these alternatives on p. 18 to read the following:

“One potential solution is the immobilization of DNA targets and reporters that would localize the collateral activities to a discrete location, such as an individual spot on a microarray or encapsulated within a water droplet.”

Applications requiring PAM-independent detection. While enabling RNA detection without PAM restrictions would generally expand the applicability of CRISPR-based diagnostics, we realized we overlooked pointing out where such an expansion would be helpful. We therefore added the following to the introduction on p. 3 when describing prior efforts to enable PAM-free RNA detection:

“These limitations hamper the detection of confined sequence differences, such as SNPs or hypervariable regions in rRNA¹⁹. The inability to directly detect RNA affects RNA biomarkers such as RNA viruses or alternative splice products, although the common use of reverse transcription and pre-amplification lessens this restriction. Nonetheless, these restrictions have driven numerous efforts to find workarounds.”

Broader context of the technology. The reviewer noted that comparisons to existing (and non-CRISPR) molecular diagnostic tests, such as costs, would provide broader relevance. We believe these comparisons are premature, as more work needs to be done to incorporate real-world demonstrations likely requiring amplification approaches such as RPA or RCA. At that point, realistic calculations can be made about costs. The reviewer also noted fluorescence as a potential cost barrier, although collateral cleavage activity by Cas12 and Cas13 nucleases have been linked to lateral flow assays (e.g., doi: 10.1126/science.aaq0179) as well as smart materials (doi: 10.1126/science.aaw5122), offering a simple and accessible readout. We now note this alternative readout with the following on p. 17 of the discussion:

“Finally, while we relied on a fluorescence-based readout, collateral cleavage by Cas12 nucleases can also be linked to other readouts, such as lateral flow assays⁸¹ or even smart materials⁸².”

REVIEWERS' COMMENTS

Reviewer #1 (Remarks to the Author):

The authors have significantly improved the manuscript by providing additional experimental data and carefully revising the manuscript. My previous concerns have been well addressed.

Reviewer #2 (Remarks to the Author):

The authors have satisfactorily addressed my concerns. The new data , especially the comparisons to DETECTR and SHERLOCK, make for a comprehensive study.

Reviewer #3 (Remarks to the Author):

The authors have adequately responded to this reviewer's suggestions and requested revisions. The revised manuscript provides a more complete comparison of the authors' newly developed PAM-less Cas12-based RNA detection system versus existing approaches while summarizing the key features that distinguish the authors' approach from other RNA sensing or quantification methods.